# FLOW-GUIDED LATENT REFINER POLICIES FOR SAFE OFFLINE REINFORCEMENT LEARNING

## ABSTRACT

Safe offline reinforcement learning remains challenging due to two coupled obstacles: (i) reconciling soft penalty designs with hard safety requirements, and (ii) avoiding out-of-distribution (OOD) actions when the learned policy departs from the behavior data. Existing approaches often rely on penalty tuning that under- or over-regularizes safety, solve constrained objectives that depend on accurate simulators or online rollouts, or train powerful generative policies that still explore low-density, safety-unknown regions at deployment. We introduce a constraint-free offline framework that addresses both issues by (a) learning a flow-based latent action manifold that concentrates density on empirically safe regions and admits tractable bounds on policy deviation and OOD shift, and (b) applying a lightweight refiner stage that performs small, ordered updates in latent space to decouple reward, safety, and OOD control, stabilizing multi-objective optimization. This design keeps policy search inside the modeled data manifold, while a feasibility-aware training signal steers the refiner toward in-support, low-violation solutions without requiring explicit constraints or online interaction. Across a range of safe offline benchmarks, the proposed method achieves lower violation rates while matching or outperforming baselines in return, demonstrating its potential as a practical and effective approach to safer offline policy learning.

## 1 INTRODUCTION

Safe offline reinforcement learning (Safe Offline RL) seeks to learn policies that maximize return while satisfying stringent safety requirements from a fixed dataset—without risky, expensive online interaction (Levine et al., 2020). Training from logs allows practitioners to leverage prior operations, simulators, or demonstrations to deploy policies in safety-critical domains (robotics (Wu et al., 2024), driving (Zhang et al., 2025), industrial control (Yu et al., 2025; Wang et al., 2025)) where exploration is untenable, offering clear practical benefits over online learning.

However, simultaneously achieving high performance and strict safety from static data remains elusive (Kushwaha et al., 2025). Most prior work (Ding & Lavaei, 2023; Le et al., 2019; Lee et al., 2022; Fujimoto et al., 2019) encodes safety as soft constraints—risk penalties or constrained Markov decision processes (CMDPs) (Altman, 2021) with Lagrangian updates, so violations are discouraged in expectation. When constraints are tight or objectives conflict, these updates often under-enforce safety, yielding policies that either ignore constraints or require brittle penalty tuning, which is undesirable in engineering settings that demand near-zero violations. Hard-constraint formulations (Wang et al., 2023; Yu et al., 2022) strengthen safety but typically induce conservatism and measurable return sacrifice, especially offline, where feasible exploration is absent.

Orthogonal to constraint design, a second challenge is out-of-distribution (OOD) shift. Offline policies must evaluate and improve without querying unseen actions; otherwise, bootstrapping on OOD actions produces extrapolation error and overestimation, which in turn elevates safety risk at deployment (Kumar et al., 2019). Recent studies further note that OOD states at test time can also degrade behavior, indicating that distribution shift is a coupled state–action phenomenon (Levine et al., 2020; Kushwaha et al., 2025). Optimizing safety and return jointly under OOD constraints is therefore difficult. A prominent line of work learns a generative manifold in the latent action or trajectory space and restricts policy updates to this manifold by using VAE-, flow-, or diffusion-based generative policies combined with latent-constraint methods (Zhou et al., 2021; Koirala et al.,

2024; Akimov et al., 2022). These approaches reduce extrapolation by biasing policies toward high-density regions of the learned model, but OOD control typically remains *implicit*—through decoder support, bounded bases, or density thresholds—and is rarely coupled with safety to yield explicit, tunable guarantees on distribution shift in safe offline RL.

To tackle these challenges, we introduce a density-first framework for safe offline RL that enforces stringent safety while simultaneously optimizing for high returns. Our approach treats safety assurance and OOD control as a representation problem within a task-conditioned latent manifold. Specifically, we equip the critics with Hamilton–Jacobi (HJ)–inspired (Bansal et al., 2017) safety signals: feasibility values are learned via a reversed expectile objective, and action-values are updated using an HJ-style backup derived from sparse safety labels, yielding reliable feasibility estimates directly from offline data. On top of this estimator, a structured conditional flow model acts as a latent prior that shapes the density so that, for each state, the induced action distribution concentrates around data-supported, empirically safe regions. Actions are generated by a high-capacity decoder that remains frozen during refinement; together with the invertible, exact-likelihood flow, this enables us to derive theoretical upper bounds on distributional shift in both the action and policy spaces, thereby offering

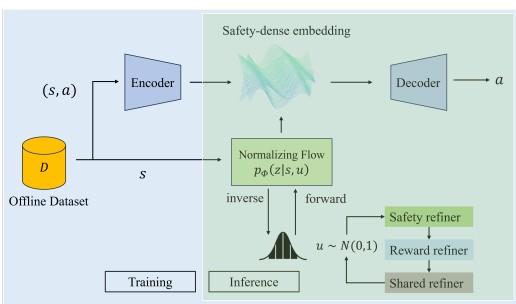

Figure 1: Overview of the proposed method. An encoder maps $(s, a)$ into a safety-dense latent embedding. A conditional normalizing flow $p_\phi(z|, u)$ with base $u \sim \mathcal{N}(0, 1)$ serves as the prior, providing exact forward/inverse transforms between the base and latent spaces; a decoder then reconstructs actions $a$ from $z$. At inference, three refiners (safety, reward, and a shared refiner) operate in the *base* Gaussian space to adjust samples toward high-density, in-support regions—maximizing return while suppressing OOD actions and enforcing safety constraints.

principled control over OOD actions in the offline setting. Building on this property, we develop a three-expert refiner—comprising reward, safety, and shared experts—that performs small, ordered updates in the base latent space with decoupled, advantage-weighted objectives. This design pushes reward improvements within feasibility-shaped regions while pulling the policy away from unsafe, low-density areas, stabilizing multi-objective optimization and tying safety, reward, and OOD suppression together under purely offline training.

Extensive experiments across diverse safe offline benchmarks demonstrate that combining safety-shaped density with latent-space refinement yields a consistently better return–safety trade-off under hard-constraint scenarios compared to strong baselines.

## 2 PRELIMINARIES

**Safe offline RL** Safe RL is typically formulated as a Constrained Markov Decision Process $\mathcal{M} = (\mathcal{S}, \mathcal{A}, T, r, h, c, \gamma)$, where $h : \mathcal{S} \to \mathbb{R}$ encodes a state constraint and $c(s) = \max\{h(s), 0\}$ is the induced cost, with $c(s) > 0$ indicating a constraint violation. The discount factor is $\gamma \in (0, 1)$. In the offline setting, we are given a fixed dataset $\mathcal{D} = \{(s, a, r, c, s', d)\}$ collected by an unknown behavior policy $\pi_\beta$. We adopt the basic offline safe RL objective:

$$\max_\pi \ \mathbb{E}_s[V_r^\pi(s)] \quad \text{s.t.} \quad \mathbb{E}_s[V_c^\pi(s)] \leq \ell, \qquad D_{\text{KL}}(\pi \parallel \pi_\beta) \leq \varepsilon, \tag{1}$$

where $V_r^\pi(s) = \mathbb{E}_\pi\big[\sum_{t=0}^\infty \gamma^t r(s_t, a_t) \mid s_0 = s\big]$ is the reward value function and $V_c^\pi(s) = \mathbb{E}_\pi\big[\sum_{t=0}^\infty \gamma^t c(s_t) \mid s_0 = s\big]$ is the cost value function. $\ell$ is the cost limit. The KL divergence $D_{\text{KL}}(\cdot\|\cdot)$ constrains the deviation of $\pi$ from the behavior policy $\pi_\beta$. In this work, we target on the zero cost budget case ($\ell = 0$); see Appendix B.2 for a discussion of non-zero budgets.

**Normalizing flows.** Normalizing flows (NFs)[1]. (Kobyzev et al., 2020) are powerful generative models for complex distribution modeling. Let $u \sim \mathcal{N}(0, I)$ and $z = f_\phi(u; \text{cond})$ be a bijection

---

[1]In this paper, 'flow' always refers to normalizing flows (e.g., RealNVP/CNF-style models), and should not be confused with flow-matching or probability flow ODE terminology.

conditioned on cond (e.g., state or task context). The log-density of $z$ is computed by the change-of-variables formula (derivation is deferred to Appendix C.1):

$$\log p_\phi(z \mid \text{cond}) = \log p(u) + \log\left|\det \frac{\partial u}{\partial z}\right|. \tag{2}$$

In our implementation, we adopt a RealNVP-style (Dinh et al., 2016) flow architecture based on coupling layers. Each layer splits the input $z$ into two parts: an identity component $z_{\text{id}}$ that remains unchanged, and a transform component $z_{\text{tr}}$ that is updated through an affine transformation:

$$z'_{\text{tr}} = z_{\text{tr}} \odot \exp s_\phi(z_{\text{id}}, \text{cond}) + t_\phi(z_{\text{id}}, \text{cond}), \qquad \log|\det J| = \sum s_\phi(\cdot), \tag{3}$$

where $s_\phi$ and $t_\phi$ are scale and translation networks. These layers yield tractable log-likelihoods and exact inverses by construction. Stacking multiple such layers increases the expressiveness of the model while preserving efficient computation due to the triangular structure of the Jacobian.

# 3 METHODOLOGY

To address the twin challenges of under-enforced soft constraints and out-of-distribution drift in offline settings, we first recast the objective as a *state-wise zero-violation* hard constraint. Concretely, we require $h(s_t) \leq 0$ almost surely under $a_t \sim \pi(\cdot \mid s_t)$ for all $t \in \mathbb{N}$. Consequently, the soft safety constraint in Eq. 1 is replaced by a state-wise requirement together with an offline trust region:

$$\max_\pi \mathbb{E}_s[V_r^\pi(s)] \quad \text{s.t.} \quad V_c^\pi(s) \leq 0, \qquad D_{\text{KL}}(\pi \| \pi_\beta) \leq \varepsilon. \tag{4}$$

Building on this formulation, we adopt a base-space refinement strategy, where optimization is performed entirely within a conditional latent density that is confined to a data-supported manifold, as illustrated in Figure 1. We refer to our method as **FLRP**—*Flow-guided Latent Refiner Policies*—which enables in-distribution policy improvement with near-zero constraint violations. The core components of FLRP are detailed in the following subsections.

## 3.1 FEASIBILITY-BASED VALUE FUNCTION

The state-wise zero-violation requirement in Eq. 4 calls for a representation that certifies safety along the entire trajectory, not only in expectation. Hamilton–Jacobi (HJ) reachability Bansal et al. (2017) from safe control provides exactly such a representation through signed safety functions and value-based certificates, and has been shown to be effective for enforcing hard constraints in recent safe RL studies (Fisac et al., 2018; Yu et al., 2022). Following this line, we cast the hard constraint into a pair of feasibility value functions based on Definition 1 that we can learn from offline data and then use as a unified signal for policy generation and refinement.

**Definition 1** (Optimal feasible value functions). *Let $h : \mathcal{S} \to \mathbb{R}$ be a signed safety function with $h(s) \leq 0$ denoting safety. The optimal state-wise and action-wise feasibility values are defined by*

$$V_h^\star(s) := \min_\pi \max_{t \in \mathbb{N}} h(s_t), \quad s_0 = s, \ a_t \sim \pi(\cdot \mid s_t), \ s_{t+1} \sim T(\cdot \mid s_t, a_t), \tag{5}$$

$$Q_h^\star(s, a) := \min_\pi \max_{t \in \mathbb{N}} h(s_t), \quad s_0 = s, \ a_0 = a, \ a_{t \geq 1} \sim \pi(\cdot \mid s_t). \tag{6}$$

By construction, $V_h^\star(s) \leq 0$ implies the existence of a policy whose entire trajectory from $s$ remains safe; likewise, $Q_h^\star(s, a) \leq 0$ certifies zero violations when starting with action $a$. In offline settings, they can be estimated by the Feasible Bellman Operator with a discounted factor $\gamma$.

**Definition 2** (Feasible Bellman operator). *For $\gamma \in (0, 1)$ and any $Q : \mathcal{S} \times \mathcal{A} \to \mathbb{R}$, the feasible Bellman operator is defined by*

$$(\mathcal{P}^\star Q)(s, a) := (1 - \gamma) h(s) + \gamma \max\{h(s), V^\star(s')\}, \quad V^\star(s') := \min_{a'} Q(s', a'). \tag{7}$$

*This operator is a $\gamma$-contraction under the sup norm and admits a unique fixed point $Q_{h,\gamma}^\star$ with $V_{h,\gamma}^\star(s) = \min_a Q_{h,\gamma}^\star(s, a)$; as $\gamma \uparrow 1$, it recovers the HJ-style values $Q_h^\star$ and $V_h^\star$ in Definition 1. Proof is deferred to Appendix C.3.*

We parameterize $(Q_h, V_h)$ with neural networks. To avoid extrapolation errors that arise from querying actions outside the data support (Fujimoto et al., 2019), we approximate $Q_h(s, \cdot)$ by reversed expectile regression and train $Q_h$ with a one-step target that uses $V_h$ in place of $\min_{a'} Q_h(s', a')$:

$$\mathcal{L}_{V_h} = \mathbb{E}_{(s,a)\sim\mathcal{D}} \left[ \rho_{\tau_h}^{\mathrm{rev}} \left( Q_h(s, a) - V_h(s) \right) \right], \tag{8}$$

$$\mathcal{L}_{Q_h} = \mathbb{E}_{(s,a,s')\sim\mathcal{D}} \left[ \left( (1 - \gamma)h(s) + \gamma \max\{h(s), V_h^{\mathrm{tgt}}(s')\} - Q_h(s, a) \right)^2 \right]. \tag{9}$$

where $\rho_\tau^{\mathrm{rev}}(u) = \left| \tau - \mathbf{1}\{u > 0\} \right| u^2$ and $V_h^{\mathrm{tgt}}$ is a slowly updated target network. The reversed expectile with $\tau_h \in (0.5, 1)$ down-weights overly optimistic $Q_h$ values and sharpens the zero level set $V_h \approx 0$, while the target network stabilizes bootstrapping.

## 3.2 Conditional Flow-based Safe Policy Generation

Rather than learning a policy directly in action space, we model a conditional latent action distribution, where high-quality samples correspond to higher density. Thus, instead of being pushed by hard constraints, safety is pulled by density: we do not project actions onto an estimated safe set, but regularize the latent actions to remain in the high-density region of the flow, while unsafe behaviours are relegated to low-density regions. Given the empirical feasibility signals learned in Sec. 3.1, we instantiate a conditional flow prior/posterior with a decoder. Compared with other generative models, normalizing flows offer exact likelihood, tractable inverse mapping, and strong expressivity (Papamakarios et al., 2021)—making them well-suited for both density modeling and OOD control.

**Safety-weighted ELBO.** Let $u \sim \mathcal{N}(0, I)$ be a base latent vector. The prior flow maps $u$ to a latent variable $z = f_\phi(u; s)$, where the log-density is tractable:

$$\log p_\phi(z \mid s) = \log p(u) + \log \left| \det \frac{\partial u}{\partial z} \right|. \tag{10}$$

The posterior flow $q_\psi(z \mid s, a)$ serves as an amortized recognizer, while a decoder $\pi_\theta(a \mid s, z)$ maps latent codes back to actions. Training follows a safety-weighted variational objective that encourages accurate reconstruction and alignment with the prior:

$$\mathcal{L}_{\mathrm{ELBO}} = \mathbb{E}_{(s,a)\sim\mathcal{D}} \mathbb{E}_{z\sim q_\psi} \left[ -w(s, a) \log \pi_\theta(a \mid s, z) \right] + \beta \, \mathbb{E}_{(s,a)\sim\mathcal{D}} \left[ w(s, a) \, \mathrm{D_{KL}} \left( q_\psi \,\|\, p_\phi \right) \right], \tag{11}$$

where $w(s, a) = \sigma(-Q_h(s, a)/T_q) \, \sigma(-V_h(s)/T_v)$ is a feasibility-weighted score derived from the critics in Sec. 3.1, $T_v$ and $T_q$ are temperatures, and $\sigma$ is the logistic function. We formally justify that the above objective remains a consistent variational estimator by showing that it performs a KL projection of the model joint distribution onto a safety-weighted behavior distribution, as stated in the following lemma.

**Lemma 1.** *Let $\tilde{p}_\mathcal{D}(s, a) \propto w(s, a) \, p_\mathcal{D}(s, a)$ be a behavior-weighted empirical distribution. Then*

$$\mathcal{L}_{ELBO} = \mathrm{const} + D_{\mathrm{KL}} \left( \tilde{p}_\mathcal{D}(s, a) \, q_\psi(z \mid s, a) \,\|\, p_\phi(z \mid s) \, \pi_\theta(a \mid s, z) \right).$$

*This result shows that $\mathcal{L}_{flow}$ amounts to a KL projection of the behavior-weighted posterior onto the generative model distribution. The proof is provided in Appendix C.3.*

**Prior Density Shaping.** Compared to a Gaussian prior, the flow-based prior is capable of modeling more complex and multimodal latent structures, but this expressiveness also introduces challenges during training. To mitigate these difficulties, we introduce a regularization objective that encourages empirically feasible regions in the action space to be mapped back to high-density regions in the latent base space. A key advantage of normalizing flows is their ability to compute an exact inverse transformation from $z$ to $u$. We leverage this to define the following prior-shaping loss:

$$\mathcal{L}_{\mathrm{shape}} = \mathbb{E}_{(s,a)\sim\mathcal{D}} \left[ \exp(Q_r(s, a) - V_r(s)/\beta_r) \cdot \mathbf{I}_{\mathrm{feas}}(s, a) \cdot \left\| T_\phi^{-1}(z_q \mid s) \right\|^2 \right] \tag{12}$$

Here, $\mathbf{I}_{\mathrm{feas}}(s, a) = \mathbf{1}\{Q_h(s, a) \leq 0\}$ is a binary indicator derived from the feasibility critic, and $T_\phi^{-1}(z_q \mid s)$ denotes the inverse transformation that maps a decoded action back to the latent base space. This encourages the flow prior to assign higher and smoother base-space density to actions that are both safe and high-reward, thereby shaping the latent manifold to better align with feasible and desirable behaviors.

**Freezing the decoder and distribution shift.** At inference time, actions are generated by sampling $u \sim \mathcal{N}(0, I)$, transforming it through the prior flow $z = f_\phi(u; s)$, and decoding via $a = \pi_\theta(z, s)$. In the subsequent refinement stage (Sec. 3.3), the decoder $\pi_\theta$ is frozen and only $u$ is updated. This confines policy updates to the safety-shaped latent manifold and avoids reintroducing distribution shift through unconstrained decoding.

We show in the following that, under a fixed decoder, the divergence between the learned policy and the behavior policy can be decomposed into controllable terms.

**Lemma 2.** *Let $\pi_0(\cdot|s) := T_{s\#}\mathcal{N}$ be the action distribution obtained by pushing the standard Gaussian through the frozen prior and decoder, and $\Pi_\theta(a \mid s)$ denotes the learned policy distribution (after refinement). Assume absolute continuity and a bounded density ratio $R_\theta(s) := \sup_a \frac{\pi_0(a|s)}{\pi_\beta(a|s)} < \infty$ on the data support. Then for any state $s$ (proofs are in Appendix C.4),*

$$D_{\mathrm{KL}}\big(\Pi_\theta(\cdot|s) \,\|\, \pi_\beta(\cdot|s)\big) \,\leq\, D_{\mathrm{KL}}\big(\Pi_\theta(\cdot|s) \,\|\, \pi_0(\cdot|s)\big) \,+\, \log R_\theta(s).$$

*Moreover, by data-processing inequality (DPI) (Beaudry & Renner, 2011) and flow invariance, $D_{\mathrm{KL}}(\Pi_\theta\|\pi_0) \leq D_{\mathrm{KL}}(q_u\|\mathcal{N})$, hence $D_{\mathrm{KL}}(\Pi_\theta\|\pi_\beta) \leq D_{\mathrm{KL}}(q_u\|\mathcal{N}) + \log R_\theta(s)$.*

This result shows that the decoder decouples policy shifts into (i) a base-space divergence term and (ii) a modeling error term, both of which can be controlled during training.

**Full objective.** We summarize the flow module's objective as:

$$\mathcal{L}_{\mathrm{flow}} = \mathcal{L}_{\mathrm{ELBO}} + \mathcal{L}_{\mathrm{shape}} + \lambda_H \big(H_0 - \mathbb{E}_{q_\psi}[-\log q_\psi(z \mid s, a)]\big)_+, \tag{13}$$

where the final term softly enforces a minimum posterior entropy to prevent mode collapse. Having shaped a structured latent manifold through feasibility-aware density modeling, we next develop a refiner module that further improves performance by optimizing within this base space.

## 3.3 BASE-SPACE OPTIMIZATION VIA EXPERT REFINER

While the flow module already shapes a safety-aware latent manifold, it does not directly optimize task performance, as a high reward is also desired. Inspired by recent progress on Mixture-of-Experts (MoE) (Jayawardana et al., 2025; Obando-Ceron et al., 2024) architectures, we design an expert refiner that operates on the Gaussian base latent $u \sim \mathcal{N}(0, I)$ learned in Sec. 3.2. The refiner performs small, ordered updates in the base space to improve reward while keeping search confined to the safety-shaped manifold.

**Architecture.** The refiner consists of three latent-space experts: a reward expert $f_r$, a safety expert $f_h$, and a shared expert $f_{\mathrm{sh}}$. Each expert applies a residual update in the latent base space conditioned on the state $s$. At each refinement step $t = 0, \ldots, T - 1$, we start from $u_0 \sim \mathcal{N}(0, I)$ and apply the following sequential updates:

$$u_{t+1} = u_t + f_k(s, u_t), \quad \text{for } k \in \{r, h, \mathrm{sh}\},$$

where the final update is always performed by the shared expert $f_{\mathrm{sh}}$. After $T$ steps, the refined latent $u_T$ is mapped to $z = f_\phi(u_T; s)$ via the frozen prior flow, and then decoded to an action distribution $\pi_\theta(\cdot \mid s, z)$ using the decoder. We denote its decoded mean by $\bar{a}(s, u_T)$ and use it for downstream evaluation or rollouts.

**Expert-specific objectives.** Let $\bar{a}(s, u_T) := \arg\max_a \pi_\theta(a \mid s, f_\phi(u_T; s))$ denote the decoded mean action, and reuse the learned critics $(Q_r, V_r)$ and $(Q_h, V_h)$ from Sec. 3.1. Each expert is trained using a modular, advantage-weighted regression (AWR) (Peng et al., 2019; Hansen-Estruch et al., 2023) objective:

*(i) Safety expert.* Minimizes the violation gap with a push–pull form:

$$L_h = \mathbb{E}_{s,a\sim\mathcal{D}}\big[\phi\big(Q_h(s, \bar{a}(s, u_T)) - V_h(s)\big) + w_h(s, a) \cdot ||\bar{a}(s, u_T) - a||_2\big], \tag{14}$$

where where $w_h(s) = \exp\big(-[Q_h(s, \bar{a}) - V_h(s)]/\beta_h\big) \cdot \mathbf{I}_{\mathrm{feas}}$, and $\phi(\cdot)$ is a soft penalty (e.g., softplus). The first term penalizes the positive safety advantage $Q_h(s, \bar{a}) - V_h(s)$ of the refined action, while the second term performs supervised regression on safety-weighted behaviour data.

*(iI) Reward expert.* Maximizes return within feasible states as a supervised learning:

$$\mathcal{L}_r = -\mathbb{E}_{s,a\sim\mathcal{D}}\left[w_r(s,a)\ \cdot\|\bar{a}(s,u_T)-a\|_2\right].\tag{15}$$

where $w_r(s,a) = \exp\big([\,Q_r(s,a)-V_r(s)\,]/\beta_r\big)\cdot\mathbf{I}_{\text{feas}}$ up-weights positive reward advantage and prevents reward-only updates from steering into unsafe states.

*(iii) Shared expert.* Regularizes refinement in the *base* space. As stated in Lemma 2: once the decoder is frozen, the policy shift is entirely induced by the divergence of the refined base distribution $D_{\text{KL}}(q_u\|\mathcal{N})$. Considering the base is a standard Gaussian distribution, we use its energy as an explicit regularizer, together with a small proximal term that discourages large steps:

$$\mathcal{L}_{\text{sh}} = \|u_T\|^2 + \|u_T-u_0\|^2.\tag{16}$$

The full refiner loss is:

$$\mathcal{L}_{\text{ref}} = \lambda_r\mathcal{L}_r + \lambda_h\mathcal{L}_h + \lambda_{\text{sh}}\mathcal{L}_{\text{sh}}\tag{17}$$

Refining in the base space with a fixed process provides distributional control for *all* downstream spaces. Because the flow and decoder are both invertible or frozen, any change in the base space deterministically propagates through the latent and action spaces. While Lemma 2 establishes a general data-processing inequality under pushforward mappings, we now apply this result specifically to our architecture. The next lemma formalizes the KL chain via pushforwards in our method.

**Lemma 3.** *Let $q_u$ be the refined base distribution and $\mathcal{N}$ the standard Gaussian. Let $f_\phi(\cdot;s)$ be the (invertible) flow and $q_z = f_{\phi\#}q_u$, $p_\phi = f_{\phi\#}\mathcal{N}$, and action distributions $\pi = T_{s\#}q_u$, $\pi_0 = T_{s\#}\mathcal{N}$ with $T_s(u) := \bar{a}(s,u)$. Then (proofs are in Appendix C.5):*

$$D_{\text{KL}}\big(\pi(\cdot\mid s)\,\|\,\pi_0(\cdot\mid s)\big)\ \le\ D_{\text{KL}}\big(q_z\,\|\,p_\phi\big)\ =\ D_{\text{KL}}\big(q_u\,\|\,\mathcal{N}\big).\tag{18}$$

The equality follows from the invariance of KL under invertible mappings (the flow), and the inequality is the data-processing inequality through the decoder.

**Corollary 1** (Deviation bounds from base KL). *Let $L_g$ be the Lipschitz constant of $g_\theta$ on the latent chart, $W_2(\cdot,\cdot)$ denotes the 2-Wasserstein distance, and $\text{TV}(\cdot,\cdot)$ stands for total variation distance between distributions. Then for any $s$ (proofs are in Appendix C.6):*

$$W_2\big(\pi,\pi_0\big) \le L_g\sqrt{2\,D_{\text{KL}}(q_u\,\|\,\mathcal{N})}$$

$$\text{TV}(\pi,\pi_\beta) \le \sqrt{\tfrac{1}{2}\,D_{\text{KL}}(\pi\,\|\,\pi_0)} + \text{TV}(\pi_0,\pi_\beta)\tag{19}$$

*and for any measurable OOD region $\mathcal{O}$:*

$$\pi(\mathcal{O}) \le \pi_\beta(\mathcal{O}) + \sqrt{\tfrac{1}{2}\,D_{\text{KL}}(q_u\,\|\,\mathcal{N})} + \text{TV}(\pi_0,\pi_\beta).\tag{20}$$

These results justify our design: keeping $D_{\text{KL}}(q_u\,\|\,\mathcal{N})$ small bounds downstream deviation—latent, action, and final policy—across multiple metrics, whereas direct perturbations in $z$ or $a$ lack such guarantees. Building on this, Appendix C.8 derives explicit reward and cost policy-gap guarantees in terms of the base-space KL upper bound and the prior–behavior mismatch. This further motivates us to optimize in the base space, where our loss concentrates mass in high-density regions so that stable base-space updates induce meaningful latent refinements.

### 3.4 PRACTICAL IMPLEMENTATION

We employ expectile regression to obtain in-sample, asymmetric value estimates that are biased toward high-value actions without querying out-of-distribution actions, following the practice in IQL (Kostrikov et al., 2021), which trains $V_r$ using asymmetric expectile regression and $Q_r$ by TD updates toward $V_r$.

$$\mathcal{L}_{V_r} = \mathbb{E}_{(s,a)\sim\mathcal{D}}\big[\rho_{\tau_r}\big(Q_r(s,a)-V_r(s)\big)\big],\quad \rho_\tau(u) = \big|\tau-\mathbf{1}\{u<0\}\big|\,u^2,\tag{21}$$

$$\mathcal{L}_{Q_r} = \mathbb{E}_{(s,a,s')\sim\mathcal{D}}\Big[\big(Q_r(s,a)-\hat{Q}_r(s,a)\big)^2\Big],\quad \hat{Q}_r(s,a) := r(s,a)+\gamma V_r(s').\tag{22}$$

Table 1: Performance Comparison on DSRL benchmark. ↑ means the higher the better, ↓ means the lower the better.

| Task | BCQL | | CPQ | | CDT | | FISOR | | LSPC | | FLRP(Ours) | |
|---|---|---|---|---|---|---|---|---|---|---|---|---|
| | reward ↑ | cost ↓ | reward ↑ | cost ↓ | reward ↑ | cost ↓ | reward ↑ | cost ↓ | reward ↑ | cost ↓ | reward ↑ | cost ↓ |
| *Safety-Gymnasium* | | | | | | | | | | | | |
| CarButton1 | 0.16 | 4.20 | 0.13 | 2.44 | 0.21 | 1.60 | -0.04 | 0.58 | -0.15 | 0.58 | 0.03 | 0.36 |
| CarButton2 | 0.07 | 3.47 | 0.17 | 7.05 | 0.13 | 1.58 | -0.01 | 0.22 | -0.03 | 0.59 | 0.04 | 0.38 |
| CarPush1 | 0.09 | 0.56 | -0.14 | 0.80 | 0.31 | 0.40 | 0.26 | 1.23 | 0.21 | 0.13 | 0.20 | 0.04 |
| CarPush2 | 0.06 | 0.61 | 0.10 | 5.66 | 0.19 | 1.30 | 0.16 | 0.71 | 0.04 | 1.37 | 0.24 | 0.36 |
| CarGoal1 | 0.13 | 0.90 | 0.22 | 0.79 | 0.66 | 1.21 | 0.42 | 0.88 | 0.23 | 0.71 | 0.27 | 0.00 |
| CarGoal2 | 0.13 | 2.38 | 0.17 | 3.10 | 0.48 | 1.25 | 0.06 | 0.06 | 0.11 | 0.50 | 0.20 | 0.28 |
| AntVel | 0.29 | 2.08 | -0.31 | 0.00 | 0.98 | 0.39 | 0.90 | 0.00 | 0.91 | 0.02 | 0.69 | 0.00 |
| HalfCheetahVel | 1.04 | 7.06 | 0.08 | 2.56 | 0.97 | 0.55 | 0.88 | 0.00 | 0.86 | 0.18 | 0.94 | 0.16 |
| SwimmerVel | 0.29 | 4.10 | 0.31 | 2.66 | 0.67 | 1.47 | 0.01 | 0.01 | 0.47 | 1.26 | 0.06 | 0.00 |
| **Safety-Gym Avg** | 0.25 | 2.82 | 0.08 | 2.78 | 0.51 | 1.08 | 0.29 | 0.40 | 0.29 | 0.59 | 0.33 | 0.18 |
| *Bullet-Safety-Gym* | | | | | | | | | | | | |
| AntRun | 0.05 | 4.63 | 0.13 | 0.01 | 0.69 | 1.24 | 0.45 | 0.76 | 0.94 | 1.46 | 0.52 | 0.00 |
| BallRun | 0.35 | 0.20 | 0.85 | 13.67 | 0.88 | 0.86 | 0.14 | 0.00 | 0.08 | 0.00 | 0.16 | 0.00 |
| CarRun | 0.75 | 2.51 | 0.75 | 0.52 | 0.99 | 1.47 | 0.80 | 0.00 | 0.75 | 0.22 | 0.87 | 0.00 |
| DroneRun | 0.65 | 0.71 | 0.26 | 0.44 | 0.71 | 0.60 | 0.41 | 0.57 | 0.62 | 1.34 | 0.59 | 0.02 |
| AntCircle | 0.61 | 1.42 | 0.00 | 0.00 | 0.46 | 2.74 | 0.23 | 0.00 | 0.40 | 0.78 | 0.45 | 0.25 |
| BallCircle | 0.79 | 1.20 | 0.40 | 4.37 | 0.79 | 1.64 | 0.45 | 0.00 | 0.29 | 1.83 | 0.46 | 0.00 |
| CarCircle | 0.64 | 1.80 | 0.49 | 4.48 | 0.70 | 1.20 | 0.34 | 0.00 | 0.28 | 0.04 | 0.66 | 0.06 |
| DroneCircle | 0.68 | 1.19 | -0.27 | 1.29 | 0.59 | 1.56 | 0.60 | 0.00 | 0.66 | 1.37 | 0.54 | 0.00 |
| **Bullet-SG Avg** | 0.57 | 1.71 | 0.33 | 3.10 | 0.73 | 1.41 | 0.43 | 0.17 | 0.50 | 0.88 | 0.54 | 0.04 |
| *Safe MetaDrive* | | | | | | | | | | | | |
| Easysparse | 0.94 | 9.25 | -0.05 | 0.15 | 0.25 | 0.15 | 0.41 | 0.50 | 0.74 | 1.55 | 0.32 | 0.20 |
| Easymean | 0.99 | 7.22 | -0.06 | 0.00 | 0.42 | 0.25 | 0.43 | 0.67 | 0.70 | 0.68 | 0.25 | 0.10 |
| Easydense | 0.20 | 1.76 | -0.06 | 0.16 | 0.35 | 1.17 | 0.52 | 1.26 | 0.74 | 1.48 | 0.33 | 0.11 |
| Mediumsparse | 0.94 | 2.83 | -0.08 | 0.12 | 0.78 | 1.24 | 0.43 | 0.08 | 0.97 | 0.79 | 0.31 | 0.06 |
| Mediummean | 0.70 | 4.45 | -0.07 | 0.16 | 0.72 | 2.74 | 0.36 | 0.02 | 0.92 | 0.89 | 0.52 | 0.63 |
| Mediumdense | 0.76 | 3.90 | -0.08 | 0.10 | 0.70 | 2.62 | 0.51 | 0.39 | 0.87 | 0.88 | 0.33 | 0.07 |
| Hardsparse | 0.49 | 3.16 | -0.05 | 0.10 | 0.26 | 0.46 | 0.33 | 0.24 | 0.52 | 1.32 | 0.35 | 0.34 |
| Hardmean | 0.29 | 3.80 | -0.05 | 0.15 | 0.20 | 0.61 | 0.27 | 0.01 | 0.41 | 0.57 | 0.28 | 0.10 |
| Harddense | 0.42 | 2.95 | -0.04 | 0.12 | 0.22 | 1.38 | 0.30 | 0.26 | 0.53 | 1.63 | 0.36 | 0.11 |
| **MetaDrive Avg** | 0.64 | 4.37 | -0.06 | 0.12 | 0.45 | 1.18 | 0.40 | 0.38 | 0.71 | 1.09 | 0.34 | 0.19 |

*Note:* **Bold**: safe policy; Gray: unsafe policy; **Bold blue**: best safe policy; **Bold**: second best safe policy

As summarized in Appendix D.5, Alg. 1, there are two main phases for the overall training procedure. In Stage 1 (critic and flow pretraining), we jointly train the safety and reward critics $(Q_h, V_h), (Q_r, V_r)$ together with the flow prior/posterior and decoder using offline transitions, and the safety-weighted ELBO and density-shaping objectives in Sec. 3.2. In Stage 2 (latent refiner training), we freeze this base model and optimize the three refiners in base space via AWR-style updates to reward and safety, together with the base-space regularizer for OOD control. All components are trained purely offline from the fixed dataset, and this two-phase, modular design lets critics, flow, and refiners specialize in feasibility shaping, density modeling, and reward–safety refinement while maintaining a consistent in-distribution optimization pipeline. At inference time, we sample $u \sim \mathcal{N}(0, I)$, apply the expert refiner for $T$ steps to obtain $u_T$, decode through the frozen flow and decoder to obtain the final action. Training details can be found in Appendix D.5.

## 4 EXPERIMENTS

**Experiment Setup.** We evaluate the proposed method against several strong offline safe RL baselines across two widely-used benchmark environments: **Safety-Gymnasium** (Ji et al., 2023), **Bullet-Safety-Gym** (Gronauer, 2022) and **Safe Metadrive** (Li et al., 2022) from the DSRL suite (Liu et al., 2023a). We adopt *normalized return* and *normalized cost* as evaluation metrics, which we refer to as "reward" and "cost" for clarity and brevity. We set a uniform cost limit of 10 for all tasks.

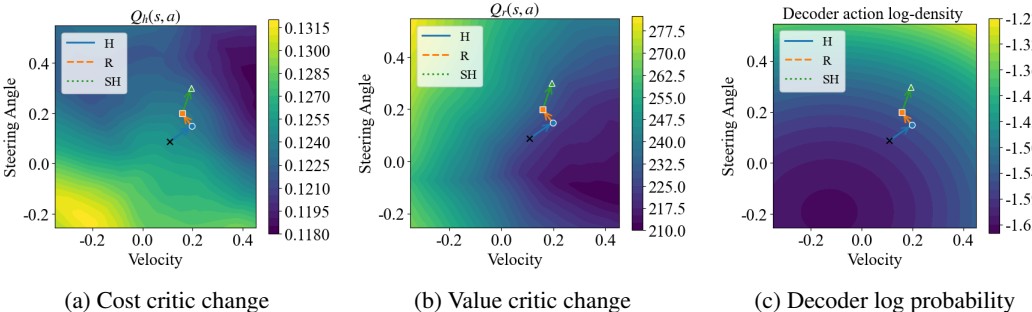

(a) Cost critic change      (b) Value critic change      (c) Decoder log probability

Figure 2: Example visualization of the refiner principle on `CarRun`. Each panel shows the 2D action space (velocity on the horizontal axis, steering angle on the vertical), where background colors indicate (a) feasibility (darker is safer), (b) reward (lighter is higher), and (c) decoder log-density (lighter is higher). The black cross is the base action from the flow prior, and colored curves (H, R, SH) show the refinement trajectories in base space, with triangles marking the final refined actions toward safer, higher-return, and data-supported regions.

**Baselines.** We compare our approach against five representative baselines: **(1) BCQL** (Fujimoto et al., 2019): A batch-constrained Q-learning with an adaptive Lagrangian penalty on constraint violations. **(2) CPQ** (Xu et al., 2022): A Q-learning methods that penalize unsafe and out-of-distribution state–action pairs. **(3) CDT** (Liu et al., 2023b): A transformer-based offline safe RL method that learns cost-conditioned action generators for constraint enforcement. **(4) LSPC** (Koirala et al., 2024): A latent safety-constrained approach that uses a conditional variational autoencoder to model safety in the latent space. **(5) FISOR** (Zheng et al., 2024): A feasibility-guided method that uses a diffusion model to policy sampling.

**Main Results** Table 1 summarizes results on Safety-Gymnasium, Bullet-Safety-Gym, and Safe MetaDrive. Overall, our method learns safe policies with competitive returns. BCQL uses a Lagrangian trade-off but often fails to meet safety constraints; CPQ is more conservative and improves safety at the cost of reward; and CDT, though capable of high returns via target conditioning, tends to violate safety more frequently. FISOR and LSPC are strong baselines with distinct characteristics. FISOR produces uniformly safe but slightly conservative policies via feasibility guidance, while LSPC is more aggressive—seeking the most rewarding action in a learned safe latent space—which can become unreliable under OOD states/actions. Our FLRP trains safety and shared refiners to concentrate probability mass in high-density regions of the encoder's latent space, naturally biasing actions toward on-support, safer choices. FLRP performs strongly on Safety-Gymnasium and Bullet-Safety-Gym, and is mildly conservative on Safe MetaDrive due to limited overlap between high-reward and low-cost regions, which complicates hard-constrained optimization. Even so, it enforces safety effectively, achieving violation rates far below the second-best method (e.g., 0.18 vs. 0.40 in Safety-Gymnasium, 0.04 vs. 0.88 in Bullet-Safety-Gym, and 0.19 vs. 0.38 in Safe MetaDrive) while maintaining strong performance.

## 5 ABLATION STUDY AND ANALYSIS

**Justification of Each Refiner.** A core challenge in safe RL is reconciling reward maximization with safety constraints, which can pull updates in opposite directions. Figure 2 illustrates this on a fixed state from the `CarRun` task. Each panel visualizes the 2D action space with velocity on the horizontal axis and steering angle on the vertical axis; the background color represents, respectively, (a) feasibility $Q_h(s,a)$ (darker is safer), (b) reward value $Q_r(s,a)$ (lighter is higher return), and (c) decoder log-density $\log \pi_\theta(a|s)$ (lighter is higher density). In this particular state, the regions associated with high reward and high safety are largely non-overlapping, and both can be misaligned with the high-density area of the action decoder. As a result, the refinement steps taken by the reward and safety refiners can diverge significantly, often steering the latent action representation into areas that are poorly supported by the decoder and thus prone to OOD issues. The shared

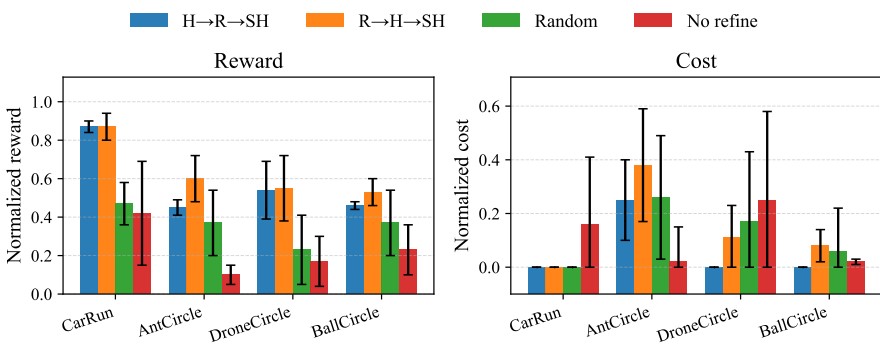

Figure 3: Effect of refiner order on normalized reward (left) and cost (right) across four tasks . Each group of bars corresponds to four refinement schedules (H→R→SH, R→H→SH, Random, No refine), with error bars showing one standard deviation.

refiner stabilizes and regularizes this process by keeping actions on support while balancing both experts, coordinating their updates when reward, safety, and data support are in tension.

**HJ-feasibility Function.** We first assess the benefit of incorporating HJ reachability by replacing the feasibility function with a cost value function. states/actions whose cost falls below the empirical 75th percentile of zero-violation samples are treated as feasible and used for flow training, while refiner training is unchanged; we denote this variant as *w/o HJ*. As reported in Table 2, this heuristic thresholding yields noisier feasibility estimates, which in turn leads to higher evaluation costs and lower returns than the HJ-based approach. In contrast, HJ reachability propagates safety constraints through the dynamics, which is robust to sampling noise and uneven cost distributions. The results indicate that structured HJ reachability is crucial for stable constraint satisfaction in offline settings.

Table 2: Ablations on HJ reachability.

| **Task** | w/o HJ | | FLRP | |
|---|---|---|---|---|
| | $r \uparrow$ | $c \downarrow$ | $r \uparrow$ | $c \downarrow$ |
| AntRun | 0.65 | 0.13 | 0.52 | 0.00 |
| BallRun | 0.08 | 0.14 | 0.16 | 0.00 |
| CarRun | 0.83 | 0.13 | 0.87 | 0.00 |
| DroneRun | 0.16 | 5.24 | 0.59 | 0.02 |
| AntCircle | 0.23 | 0.01 | 0.45 | 0.25 |
| BallCircle | 0.44 | 0.00 | 0.46 | 0.00 |
| CarCircle | 0.63 | 0.49 | 0.66 | 0.06 |
| DroneCircle | 0.56 | 0.67 | 0.54 | 0.00 |

**The Order of Refinement.** We compare four refiner schedules on four tasks (`BallCircle`, `CarRun`, `AntCircle`, `DroneCircle`) to assess how sensitive FLRP is to the refiner order: two fixed orders (H→R→SH and R→H→SH), a random permutation, and a "No refine" baseline that samples directly from the flow prior. The results in Figure 3 show that all refiner variants substantially improve normalized return over No refine, confirming the benefit of latent refinement. Across all tasks, H→R→SH and R→H→SH achieve clearly higher return than no refinement baseline with low normalized cost, while the random-order variant is intermediate but with larger variability. We also observe a consistent trade-off pattern: H→R→SH generally yields lower cost with strong but slightly lower return, whereas R→H→SH attains the highest return at the price of higher cost. This supports our design choice of using a fixed schedule with the shared refiner applied last so that it can consistently regularize and coordinate the preceding safety and reward updates.

**Other Ablations.** We further examine the effect of the prior. As a comparison, we train a variant that replaces our flow-based prior with a conventional Gaussian prior and report results in Table 3. The flow prior consistently yields higher returns and lower costs. We also study the number of refinement steps $T$ at inference on `CarCircle`. We do not vary the refinement order: the safety expert is always applied first, and the shared expert last. This design choice reflects our latent geometry—density concentrates on safety rather than reward—so an early safety refinement helps place trajectories in high-density (feasible) regions. The intermediate refiners alternate between safety and reward experts. As shown in Figure 4, increasing $T$ reduces cost and variability: a larger $T$ is more likely to explore the learned latent space and lowers the rate of out-of-distribution actions. The trade-off is that a very large $T$ can induce slightly more conservative behavior. In practice, an intermediate value (e.g., $T = 3$) can yield a favorable trade-off.

Table 3: Ablations on the prior used.

| Task | Gaussian Prior | | Flow Prior | |
|---|---|---|---|---|
| | $r \uparrow$ | $c \downarrow$ | $r \uparrow$ | $c \downarrow$ |
| CarButton1 | -0.14 | 0.22 | 0.03 | 0.36 |
| CarButton2 | 0.01 | 0.82 | 0.04 | 0.38 |
| CarPush1 | 0.07 | 0.08 | 0.20 | 0.04 |
| CarPush2 | 0.06 | 0.00 | 0.24 | 0.36 |
| CarGoal1 | 0.06 | 0.00 | 0.27 | 0.00 |
| CarGoal2 | 0.05 | 0.74 | 0.20 | 0.28 |

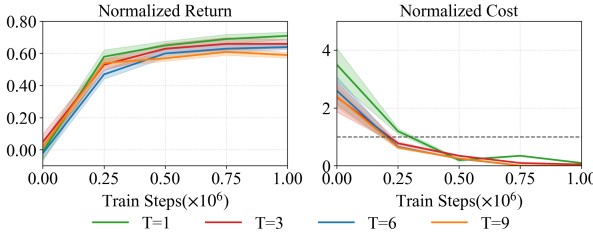

Figure 4: Ablation on the number of refinement steps.

Table 4: Representative generative latent(-space) policy methods for offline (safe) RL.

| Method | Backbone | Safety-aware? | Likelihood | OOD control |
|---|---|---|---|---|
| PLAS (Zhou et al., 2021) | CVAE | No | Approx. | Implicit (latent manifold) |
| LSPC (Koirala et al., 2024) | CVAE | Yes | Approx. | Implicit (bounded latent) |
| LDGC (Venkatraman et al., 2023) | Diffusion | No | Implicit | Implicit (batch-constrained) |
| FISOR (Zheng et al., 2024) | Diffusion | Yes | Implicit | Implicit (HJ-weighted data) |
| CNF (Akimov et al., 2022) | Flow | No | Exact | Implicit (bounded latent) |
| **FLRP (ours)** | **Flow** | **Yes** | **Exact** | **Explicit (base-KL)** |

## 6 RELATED WORK

Offline safe RL aims to learn constraint-satisfying policies from fixed datasets, avoiding risky online interaction. Early work incorporates penalty or Lagrangian terms into value learning—e.g., CPQ (Xu et al., 2022), BCQ-Lag (Fujimoto et al., 2019), and BEAR-Lag (Liu et al., 2023a)—to account for safety in the Bellman objectives. Others adopt distribution-correction methods such as COptiDICE (Lee et al., 2022), which model the stationary distribution under constraints. Sequence models like CDT (Liu et al., 2023b) and SaFormer (Zhang et al., 2023b) encode safety via cost-aware conditioning in Decision Transformer frameworks. These methods typically enforce soft constraints, allowing for occasional violations. Recent approaches (Yu et al., 2022; Ganai et al., 2023) instead leverage Hamilton–Jacobi (HJ) reachability to enforce strict state-wise safety. Complementary to these formulations, another line of work learns a generative policy or latent manifold to encourage safe behavior. Notably, LSPC (Koirala et al., 2024) learns a cost-sensitive latent policy via a CVAE prior, while FISOR (Zheng et al., 2024) couples diffusion-based behavior learning with HJ-based feasibility guidance. While these methods achieve strong empirical safety, they typically handle OOD generalization implicitly by relying on the expressivity of the generative prior or support-based constraints, without substantial improvements over general offline RL methods (Zhou et al., 2021; Akimov et al., 2022; Chen et al., 2022) in OOD robustness. Table 4 summarizes representative generative approaches and compares the key distinctions along four axes, with an extended discussion on B.1. FLRP unifies these two lines by combining a flow-based latent policy with explicit base-space KL control, and by using HJ reachability not as an external filter but as a feasibility-guided signal that shapes the latent manifold, yielding provable bounds on total variation.

## 7 CONCLUSION

We present a safe offline RL framework based on latent refinement. A multi-expert policy iteratively adjusts a base latent via safety- and reward-guided residuals, while a normalizing-flow prior shapes a feasible latent manifold for explicit safety control. We prove order-agnostic bounds on the final policy distribution and show strong performance across three standard safe RL benchmarks. The main limitations arise from the feasibility critics. The offline feasibility critics use a Hamilton–Jacobi–style Bellman operator with sparse cost, which can over-conservatively estimate value; genuinely safe but rare samples may be undervalued, introducing bias or sample inefficiency. Latent-space refinement also adds hyperparameters (e.g., expert loss weights and prior shaping temperature). That said, we used a single configuration across 26 tasks, suggesting reasonable robustness. Future work includes adaptive refinement schedules, more principled objectives for shaping the prior, and hierarchical expert architectures to improve flexibility and generalization.

**Ethics Statement.** This research does not involve human subjects, sensitive data, or practices that pose foreseeable harm. Our methodology builds upon well-established safe offline reinforcement learning benchmarks and standard datasets that are publicly available. All experiments were conducted in simulation environments with no real-world deployment or safety risk. We have made efforts to ensure transparency and reproducibility by providing code and detailed algorithmic descriptions. We adhere to the ICLR Code of Ethics, and this work upholds responsible stewardship and scientific integrity throughout.

**Reproducibility Statement.** We have taken several steps to ensure the reproducibility of our work. All theoretical results, including key lemmas and corollaries, are presented with complete assumptions and detailed proofs in the appendix. Additional implementation details, including dataset setup, training pipeline, and evaluation protocol, are also provided in the appendix. We also include an anonymous link to our core source code at: `https://anonymous.4open.science/r/FLRP-9776/`

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

## A  LLM USAGE

The authors used large language models (LLMs), specifically ChatGPT (GPT-4), solely as a language editing assistant. The LLM was employed only for grammar correction, stylistic improvements, and minor clarity revisions of the authors' own writing.

All ideas, algorithms, experimental designs, theoretical proofs, and scientific contributions presented in this paper are the sole work of the authors. The authors take full responsibility for the technical content and claims made in the paper. No content was generated or suggested by the LLM regarding methodology, experiments, or results.

## B  EXTENDED DISCUSSIONS ON RELATED WORKS

### B.1  GENERATIVE LATENT-SPACE OFFLINE RL METHODS

A growing line of offline RL methods learns policies in a low-dimensional latent action or trajectory space induced by a generative model. These approaches typically fit a conditional generative model on offline data and optimize a latent policy whose outputs are decoded back to actions, thereby constraining policy search to a data-supported manifold and reducing OOD actions. PLAS (Zhou et al., 2021) and its CVAE-based extensions, such as LAPO (Chen et al., 2022) and ELAPSE (Han et al., 2025), enhance this framework by shaping the latent distribution to emphasize high-return behaviors and mitigate collapse. In the model-based setting, C-LAP (Alles et al., 2024) learns a latent action state-space model and constrains imagined rollouts to remain within the latent prior, providing implicit conservatism. Latent diffusion approaches (Venkatraman et al., 2023) extend this idea to trajectory-level latent spaces, enabling policy optimization over semantically structured latent trajectories. Flow-based generative policies have also been explored; CNF (Akimov et al., 2022) trains a normalizing flow over actions and reduces OOD actions by bounding the base distribution under a frozen decoder. CPED (Zhang et al., 2023a) explicitly estimates the behavior density using a flow-GAN and constrains policy updates within high-density regions. For safe offline RL, LSPC (Koirala et al., 2024) encodes latent safety constraints with a CVAE and regularizes the latent policy using a safety critic, though it still relies on ELBO training and support-based constraints.

Compared with these generative latent(-space) policy methods, FLRP differs along four complementary dimensions, summarized here and in Table 4 in the main text.

1. **Task scope and safety objective.** Prior flow-based methods such as CNF (Akimov et al., 2022) do not target safe offline RL. FLRP lies in the same flow-based family but is instantiated for hard-constrained safe offline RL with near-zero violation, rather than for unconstrained or budgeted objectives.

2. **Generative backbone and likelihood.** CVAE-, flow-, and diffusion-based policies all exploit latent manifolds, but flows are invertible and admit exact likelihoods. With a Gaussian base, FLRP can monitor a base-space KL divergence and propagate it into bounds on action/policy deviation (TV/$W_2$) and OOD mass, providing a quantified, tunable notion of conservatism not available to ELBO-trained CVAEs (Zhou et al., 2021; Chen et al., 2022; Han et al., 2025) or multi-step latent diffusion models (Venkatraman et al., 2023).

3. **OOD-control mechanism.** CNF reduces OOD by making the flow's base uniform-bounded and freezing the decoder (Akimov et al., 2022), but does not explicitly control policy deviation. FLRP instead (a) retains a Gaussian base with an explicit OOD/shift bound from the base-space KL and (b) performs feasibility-guided density shaping on the base (using the flow's inverse). Together, this makes conservatism measurable and controllable, while keeping policy search within empirically safe, high-density regions.

4. **Training and inference protocol.** Safe offline RL couples reward, safety, and OOD control. Instead of relying on a single entangled loss, FLRP employs ordered small-step refiners in the base space with a frozen decoder—Safety → Reward → Shared—so updates remain in-support and non-expansive. This protocol tightly links safety, reward, and OOD suppression, exposes a clear trade-off handle, and avoids the instability of lumping all terms into one gradient..

## B.2 ADDITIONAL DISCUSSION ON HARD AND SOFT CONSTRAINT

**Hard vs. soft formulations.**  In safe RL, a *hard* (state-wise) safety constraint requires that the policy never leaves the safe set. Let $h : \mathcal{S} \to \mathbb{R}$ encodes a state constraint and $c(s) = \max\{h(s), 0\}$ is the induced cost. A hard constraint enforces

$$h(s_t) \le 0, \quad a_t \sim \pi(\cdot \mid s_t), \ \forall t \in \mathbb{N}, \tag{23}$$

which can equivalently be written as a zero-violation cost condition

$$c(s_t) = 0, \quad a_t \sim \pi(\cdot \mid s_t), \ \forall t \in \mathbb{N}. \tag{24}$$

By contrast, *soft* or *budgeted* constraints are typically expressed at the level of expected cumulative cost. Given a cost limit $l > 0$, the constraint is

$$\mathbb{E}_{\tau \sim \pi} \Big[ \sum_{t=0}^{\infty} c(s_t) \Big] \le l \quad \text{or} \quad \mathbb{E}_{\tau \sim \pi} \Big[ \sum_{t=0}^{\infty} \gamma^t c(s_t) \Big] \le l, \tag{25}$$

 and the policy is allowed to incur nonzero instantaneous violations as long as the long-term budget is respected. Recent work further extends this perspective to *real-time* budgeted safety, where the agent must adapt to dynamically specified cost budgets in the offline setting (Lin et al., 2023), as well as to risk- and distributionally-robust variants (Chow et al., 2019; Kushwaha et al., 2025).

**Design philosophies and use cases.**  Hard and soft formulations reflect different safety philosophies rather than a strict ordering of capability. Hard/near-zero-violation methods (Fisac et al., 2018; Yu et al., 2022; Zheng et al., 2024; Zhao et al., 2023) target scenarios where every violation corresponds to an unacceptable safety breach (e.g., collisions, irreversible damage, or regulatory violations); here, the emphasis is on characterizing and staying inside the feasible region. Budgeted or soft methods (Le et al., 2019; Lee et al., 2022; Liu et al., 2023b), in contrast, model cost as an allocatable resource: small, occasional violations are acceptable if they enable substantially better task performance, which is appropriate for risk-sensitive but non-safety-critical domains or applications with tunable risk budgets.

Our framework intentionally follows the hard / near-zero-violation viewpoint: we are interested in safe offline RL settings where violations correspond to genuine safety failures, and thus focus on maximizing return while keeping state-wise safety rates close to $100\%$. We view budgeted-safety approaches as complementary rather than competing; in principle, similar generative latent-space and flow-based techniques could be adapted to budgeted formulations by conditioning critics and refiners on a dynamic cost budget, which we leave as an interesting direction for future work.

## C THEORETICAL ANALYSIS

In this section, we provide the missing proofs for the theoretical results to support or validate the proposed method.

### C.1 DERIVATION OF THE FLOW DENSITY

Normalizing flows model complex distributions by transporting samples from a simple base density through an invertible transformation. In the conditional setting, let $u \sim p_0(u)$ denote a latent variable drawn from a base distribution, typically $\mathcal{N}(0, I)$, and define $z = f_\phi(u; \mathrm{cond})$, where $f_\phi(\cdot; \mathrm{cond})$ is a bijective mapping parameterized by $\phi$ and conditioned on an external variable $\mathrm{cond}$ (e.g., a state or context).

Because the map is invertible for fixed $\mathrm{cond}$, the inverse $u = f_\phi^{-1}(z; \mathrm{cond})$ is well defined. To obtain the conditional density $p_\phi(z \mid \mathrm{cond})$, we apply the change-of-variables formula for differentiable bijections:

$$p(z) = p_0\big(f^{-1}(z)\big) \cdot \left| \det \frac{\partial f^{-1}(z)}{\partial z} \right| = p_0(u) \cdot \left| \det \frac{\partial f(u)}{\partial u} \right|^{-1}, \tag{26}$$

where the second equality follows from the inverse function theorem. In our conditional setting we thus have

$$p_\phi(z \mid \text{cond}) = p_0(u) \cdot \left| \det \frac{\partial u}{\partial z} \right|, \quad u = f_\phi^{-1}(z; \text{cond}). \tag{27}$$

Using $\frac{\partial u}{\partial z} = \left( \frac{\partial z}{\partial u} \right)^{-1}$, we can express the inverse Jacobian in terms of the forward transformation:

$$\left| \det \frac{\partial u}{\partial z} \right| = \left| \det \frac{\partial f_\phi(u; \text{cond})}{\partial u} \right|^{-1}. \tag{28}$$

Substituting this identity back into the density expression gives

$$p_\phi(z \mid \text{cond}) = p_0(u) \cdot \left| \det \frac{\partial f_\phi(u; \text{cond})}{\partial u} \right|^{-1}, \quad u = f_\phi^{-1}(z; \text{cond}). \tag{29}$$

Taking logarithms yields the exact log-likelihood of the conditional flow:

$$\log p_\phi(z \mid \text{cond}) = \log p_0(u) + \log \left| \det \frac{\partial u}{\partial z} \right|, \quad u = f_\phi^{-1}(z; \text{cond}), \tag{2}$$

which corresponds to Eq. 2 in the main text.

In practice, the Jacobian determinant is computed analytically using affine coupling layers, whose triangular structure reduces the log-determinant to a sum of layerwise log-scale outputs. This makes the likelihood term efficient to compute while preserving the exactness afforded by the invertibility of the flow.

When the transformation is a composition of $L$ conditional bijections,

$$u_0 \sim p_0, \quad u_\ell = f_\ell(u_{\ell-1}; \text{cond}), \ \ell = 1, \ldots, L, \quad z = u_L, \tag{30}$$

The change-of-variables formula yields

$$\log p_\phi(z \mid \text{cond}) = \log p_0(u_0) + \sum_{\ell=1}^{L} \log \left| \det \frac{\partial u_{\ell-1}}{\partial u_\ell} \right|, \tag{31}$$

where each term uses the inverse Jacobian of layer $f_\ell$. Equivalently, this can be written as the negative sum of forward log-determinants,

$$\log p_\phi(z \mid \text{cond}) = \log p_0(u_0) - \sum_{\ell=1}^{L} \log \left| \det \frac{\partial f_\ell(u_{\ell-1}; \text{cond})}{\partial u_{\ell-1}} \right|, \tag{32}$$

which is the form implemented in practice when accumulating the density term across multiple flow layers.

### C.2 PROOF OF DEFINITION 2.

For a fixed $\gamma \in (0, 1)$ and we define $V_i(s) := \min_a Q_i(s, a)$ for $i \in \{1, 2\}$. Then for any $(s, a)$,

$$\left| (\mathcal{P}^\star Q_1)(s, a) - (\mathcal{P}^\star Q_2)(s, a) \right| = \gamma \left| \mathbb{E}_{s'} \left[ \max\{h(s), V_1(s')\} - \max\{h(s), V_2(s')\} \right] \right|$$
$$\leq \gamma \, \mathbb{E}_{s'} \left| V_1(s') - V_2(s') \right|. \tag{33}$$

Since $V_i(s') = \min_{a'} Q_i(s', a')$ and the pointwise min is 1-Lipschitz, $\left| V_1(s') - V_2(s') \right| \leq \sup_{a'} |Q_1(s', a') - Q_2(s', a')| \leq \|Q_1 - Q_2\|_\infty$. Taking the supremum over $(s, a)$ yields

$$\|\mathcal{P}^\star Q_1 - \mathcal{P}^\star Q_2\|_\infty \leq \gamma \|Q_1 - Q_2\|_\infty, \tag{34}$$

so $\mathcal{P}^\star$ is a $\gamma$-contraction under the sup norm. By Banach's fixed-point theorem, there exists a unique fixed point $Q_{h,\gamma}^\star$ and we set $V_{h,\gamma}^\star(s) := \min_a Q_{h,\gamma}^\star(s, a)$.

To connect to the undiscounted HJ-style values, assume $h$ is bounded. Let $\gamma_n \uparrow 1$ and consider the fixed points $Q^\star_{h,\gamma_n}$. Because $\{Q^\star_{h,\gamma_n}\}_n$ is uniformly bounded and $\mathcal{P}^\star$ is continuous in $\gamma$, any limit point $Q^\dagger$ satisfies, for all $(s,a)$,

$$
\begin{aligned}
Q^\dagger(s,a) &= \lim_{n \to \infty} \Big[ (1 - \gamma_n) h(s) + \gamma_n \, \mathbb{E}_{s'} \big[ \max\{h(s), \min_{a'} Q^\star_{h,\gamma_n}(s',a')\} \big] \Big] \\
&= \mathbb{E}_{s'} \big[ \max\{h(s), \min_{a'} Q^\dagger(s',a')\} \big].
\end{aligned}
\tag{35}
$$

This is the dynamic programming equation for the HJ-style (statewise zero-violation) feasibility values; hence $Q^\dagger = Q^\star_h$ and $V^\dagger = \min_a Q^\dagger(\cdot, a) = V^\star_h$. Therefore $Q^\star_{h,\gamma} \to Q^\star_h$ and $V^\star_{h,\gamma} \to V^\star_h$ as $\gamma \uparrow 1$. $\qquad\square$

## C.3 PROOF OF LEMMA 1.

Recall the weighted objective in full form:

$$
\begin{aligned}
\mathcal{L}_{\text{flow}} = {}& \mathbb{E}_{(s,a)\sim p_\mathcal{D}} \Big[ w(s,a) \, \mathbb{E}_{z \sim q_\psi(z|s,a)} \big[ -\log \pi_\theta(a \mid s,z) \big] \Big] \\
& + \beta \, \mathbb{E}_{(s,a)\sim p_\mathcal{D}} \Big[ w(s,a) \, D_{\text{KL}}\big( q_\psi(\cdot \mid s,a) \,\big\|\, p_\phi(\cdot \mid s) \big) \Big].
\end{aligned}
\tag{36}
$$

and we define the behavior-weighted data distribution $\tilde{p}_\mathcal{D}(s,a) := w(s,a) \, p_\mathcal{D}(s,a)/Z$ with normalizer $Z = \mathbb{E}_{p_\mathcal{D}}[w(s,a)]$ (a constant independent of $(\phi, \psi, \theta)$). For clarity, first consider $\beta = 1$; we return to $\beta \neq 1$ at the end. Then, up to the positive constant factor $Z$,

$$
\mathcal{L}_{\text{flow}} = Z \cdot \mathbb{E}_{(s,a)\sim \tilde{p}_\mathcal{D}} \Big\{ \mathbb{E}_{z \sim q_\psi} \big[ -\log \pi_\theta(a \mid s,z) \big] + D_{\text{KL}}\big( q_\psi \,\|\, p_\phi \big) \Big\}.
$$

Expand the KL term inside the expectation:

$$
\begin{aligned}
\mathbb{E}_{z \sim q_\psi} \big[ -\log \pi_\theta(a \mid s,z) \big] &+ \mathbb{E}_{z \sim q_\psi} \big[ \log q_\psi(z \mid s,a) - \log p_\phi(z \mid s) \big] \\
&= \mathbb{E}_{z \sim q_\psi} \Big[ \log \frac{q_\psi(z \mid s,a)}{p_\phi(z \mid s) \, \pi_\theta(a \mid s,z)} \Big].
\end{aligned}
\tag{37}
$$

Taking the expectation over $(s,a) \sim \tilde{p}_\mathcal{D}$ yields

$$
\begin{aligned}
\frac{1}{Z} \mathcal{L}_{\text{flow}} &= \mathbb{E}_{(s,a)\sim \tilde{p}_\mathcal{D}} \mathbb{E}_{z \sim q_\psi} \Big[ \log \frac{\tilde{p}_\mathcal{D}(s,a) \, q_\psi(z \mid s,a)}{\tilde{p}_\mathcal{D}(s,a) \, p_\phi(z \mid s) \, \pi_\theta(a \mid s,z)} \Big] \\
&= D_{\text{KL}}\Big( \tilde{p}_\mathcal{D}(s,a) \, q_\psi(z \mid s,a) \,\big\|\, \tilde{p}_\mathcal{D}(s,a) \, p_\phi(z \mid s) \, \pi_\theta(a \mid s,z) \Big).
\end{aligned}
\tag{38}
$$

Finally, use the identity $D_{\text{KL}}(P \| C \cdot Q) = D_{\text{KL}}(P \| Q) - \mathbb{E}_P[\log C]$ for a positive constant density factor $C$ that does not depend on the model parameters $(\phi, \psi, \theta)$; here $C = \tilde{p}_\mathcal{D}(s,a)$. Therefore,

$$
\mathcal{L}_{\text{flow}} = \text{const} \; + \; D_{\text{KL}}\Big( \tilde{p}_\mathcal{D}(s,a) \, q_\psi(z \mid s,a) \,\big\|\, p_\phi(z \mid s) \, \pi_\theta(a \mid s,z) \Big),
\tag{39}
$$

where the constant depends only on $\tilde{p}_\mathcal{D}$ (hence on $w$ and the dataset) and not on $(\phi, \psi, \theta)$. This proves the claim for $\beta = 1$.

*Extension to $\beta \neq 1$.* For a general $\beta > 0$, the same algebra shows that

$$
\mathcal{L}_{\text{flow}} = \text{const} \; + \; D_{\text{KL}}\Big( \tilde{p}_\mathcal{D}(s,a) \, q_\psi(z \mid s,a) \,\big\|\, p_\phi^{(\beta)}(z \mid s) \, \pi_\theta(a \mid s,z) \Big),
\tag{40}
$$

with a *temperature-adjusted prior* $p_\phi^{(\beta)}(z \mid s) \propto p_\phi(z \mid s)^\beta$ (i.e., the energy scaled by $\beta$). Equivalently, if one wishes to keep $p_\phi$ unchanged, absorb $\beta$ by rescaling the KL term or by introducing a decoder temperature; both formulations are strictly equivalent up to a parameter-independent constant. $\qquad\square$

### C.4 PROOF OF LEMMA 2

Let $p := \Pi_\theta(\cdot|s)$, $r := \pi_0(\cdot|s)$, $q := \pi_\beta(\cdot|s)$ w.r.t. a common dominating measure. By the elementary inequality (chain rule with a bounded density ratio)

$$D_{\mathrm{KL}}(p\|q) = D_{\mathrm{KL}}(p\|r) + \mathbb{E}_p\big[\log \tfrac{r}{q}\big] \le D_{\mathrm{KL}}(p\|r) + \log \sup_a \frac{r(a)}{q(a)} = D_{\mathrm{KL}}(p\|r) + \log R_\theta(s). \quad (41)$$

Under a frozen decoder $T_s : \mathcal{U} \to \mathcal{A}$, we treat the transformation from base latent $u$ to action $a$ as a measurable pushforward mapping. Let $q_u$ be the refined base distribution and $\mathcal{N}$ the standard Gaussian. Then the induced action distributions satisfy

$$D_{\mathrm{KL}}(T_{s\#}q_u \,\|\, T_{s\#}\mathcal{N}) \le \mathrm{KL}(q_u\|\mathcal{N}), \quad (42)$$

by the data-processing inequality (DPI) for Kullback–Leibler divergence under measurable maps; (e.g.,see Csiszár & Shields (2004)). This result states that any deterministic or stochastic channel (here, the frozen decoder $T_s$) cannot increase KL divergence. $\qquad\square$

### C.5 PROOF OF LEMMA 3

Let $f_\phi : \mathbb{R}^d \to \mathbb{R}^d$ be a smooth bijection (the prior flow). Define $q_z = f_{\phi\#}q_u$ and $p_\phi = f_{\phi\#}\mathcal{N}$. By the change-of-variables formula,

$$q_z(z) = q_u(u)\Big|\det \frac{\partial u}{\partial z}\Big|, \qquad p_\phi(z) = \mathcal{N}(u)\Big|\det \frac{\partial u}{\partial z}\Big|, \quad z = f_\phi(u). \quad (43)$$

Hence

$$D_{\mathrm{KL}}(q_z\|p_\phi) = \int q_z(z) \log \frac{q_z(z)}{p_\phi(z)} \, \mathrm{d}z = \int q_u(u) \log \frac{q_u(u)}{\mathcal{N}(u)} \, \mathrm{d}u = D_{\mathrm{KL}}(q_u\|\mathcal{N}), \quad (44)$$

i.e., KL is invariant under the bijection $f_\phi$.

Let $T_s : \mathbb{R}^d \to \mathcal{A}$ be the deterministic decoder mapping (e.g., decoded mean) with frozen $\theta$. The data-processing inequality for $f$-divergences (including KL) under a measurable pushforward gives

$$D_{\mathrm{KL}}\big(T_{s\#}q_z \,\big\|\, T_{s\#}p_\phi\big) \le D_{\mathrm{KL}}(q_z\|p_\phi). \quad (45)$$

With $\pi = T_{s\#}q_u = T_{s\#}q_z$ and $\pi_0 = T_{s\#}\mathcal{N} = T_{s\#}p_\phi$, we obtain $D_{\mathrm{KL}}(\pi\|\pi_0) \le \mathrm{KL}(q_z\|p_\phi) = D_{\mathrm{KL}}(q_u\|\mathcal{N})$, which proves Eq. 18.

### C.6 PROOF OF COROLLARY 1

For the Wasserstein bound, write $\pi = T_{s\#}q_z$ and $\pi_0 = T_{s\#}p_\phi$ with $T_s = g_\theta(\cdot, s)$. If $g_\theta$ is $L_g$-Lipschitz on the latent chart, then the pushforward is $L_g$-Lipschitz in $W_2$:

$$W_2(\pi, \pi_0) \le L_g \, W_2(q_z, p_\phi). \quad (46)$$

By Talagrand's $T_2$ inequality (Gaussian reference or log-Sobolev under mild conditions) (Otto & Villani, 2000), $W_2(q_z, p_\phi) \le \sqrt{2\, D_{\mathrm{KL}}(q_z\|p_\phi)}$, and Lemma 3 implies $W_2(\pi, \pi_0) \le L_g\sqrt{2\, D_{\mathrm{KL}}(q_u\|\mathcal{N})}$.

For total variation (TV) and OOD probability, the triangle inequality yields $\mathrm{TV}(\pi, \pi_\beta) \le \mathrm{TV}(\pi, \pi_0) + \mathrm{TV}(\pi_0, \pi_\beta)$. Pinsker's inequality (Csiszár & Shields, 2004) gives $\mathrm{TV}(\pi, \pi_0) \le \sqrt{\tfrac{1}{2} D_{\mathrm{KL}}(\pi\|\pi_0)} \le \sqrt{\tfrac{1}{2} D_{\mathrm{KL}}(q_u\|\mathcal{N})}$, using Lemma 3. For any measurable $\mathcal{O}$,

$$\pi(\mathcal{O}) - \pi_\beta(\mathcal{O}) \le \mathrm{TV}(\pi, \pi_\beta) \le \sqrt{\tfrac{1}{2} D_{\mathrm{KL}}(q_u\|\mathcal{N})} + \mathrm{TV}(\pi_0, \pi_\beta). \quad (47)$$

Rearranging completes the proof.

**Remark.** The Wasserstein bound in Corollary 1 relies on the Lipschitz continuity of the decoder $g_\theta$ with constant $L_g$. We note that this is a mild and practically enforceable assumption. During training, the decoder's Lipschitz constant can be implicitly constrained through techniques such as weight normalization Salimans & Kingma (2016), spectral normalization Miyato et al. (2018), or the gradient penalty Gulrajani et al. (2017), which are commonly used in generative modelling to enhance stability and generalization. Consequently, the theoretical bounds derived herein are not only sound but also practically relevant, as the key quantity $D_{\mathrm{KL}}(q_u|\mathcal{N})$ remains the primary lever for controlling distributional shift.

### C.7 ORDER-AGNOSTIC BOUNDS FOR SEQUENTIAL REFINEMENT

We formalize that the KL/Wasserstein/TV bounds in Lemma 3 and Corollary 1 are independent of the update order used by the experts.

**Proposition 1** (Order-agnosticity of base-space bounds). *Let $R$ be any (possibly stochastic) measurable refinement operator on the base space that maps the standard Gaussian $\mathcal{N}$ to a refined distribution $q_u = R(\mathcal{N})$, obtained by any composition/order of expert updates (e.g., Gauss–Seidel, Jacobi, interleaved mini-steps) subject to a trust region $\|u_T - u_0\| \leq \rho$. With the prior flow $f_\phi$ and decoder $g_\theta$ fixed (as in Sec. 3.2), define $\pi = T_{s\#} q_u$ and $\pi_0 = T_{s\#} \mathcal{N}$ where $T_s(u) = g_\theta(f_\phi(u; s), s)$. Then the conclusions of Lemma 3 and Corollary 1 hold verbatim with this $q_u$:*

$$
\begin{aligned}
D_{\mathrm{KL}}\big(\pi(\cdot \mid s) \,\big\|\, \pi_0(\cdot \mid s)\big) &\leq D_{\mathrm{KL}}\big(q_u(\cdot \mid s) \,\big\|\, \mathcal{N}\big), \\
W_2\big(\pi(\cdot \mid s), \pi_0(\cdot \mid s)\big) &\leq L_g \sqrt{2\, D_{\mathrm{KL}}(q_u \| \mathcal{N})},
\end{aligned}
\tag{48}
$$

*and the TV/Pinsker OOD bound remains unchanged.*

*Proof.* The proofs of Lemma 3 and Corollary 1 use only: (i) invariance of KL under the bijection $f_\phi$; (ii) data-processing for pushforwards through the frozen decoder $g_\theta$; (iii) Talagrand/Pinsker inequalities. None of these depend on the *path* that produces $q_u$, only on the *resulting* distribution $q_u$. Any expert ordering defines a measurable map whose pushforward of $\mathcal{N}$ is $q_u$; substituting this $q_u$ into the same steps yields the stated bounds. The optional trust region ensures KL finiteness and well-definedness but does not affect order independence. □

### C.8 POLICY GAP AND COMPARISON WITH PRIOR PERFORMANCE BOUNDS

We first derive simple performance bounds that relate the reward and cost gaps between the refined policy and the flow prior / behavior policy to the base-space KL regularizer used in FLRP.

**Preliminaries.** Let $\pi$ denote the final refined policy, $\pi_0$ the flow prior policy, and $\pi_\beta$ the behavior policy. Rewards and costs are bounded as $|r(s,a)| \leq R_{\max}$ and $|c(s,a)| \leq C_{\max}$. We write $J_r(\pi) := \mathbb{E}[\sum_{t \geq 0} \gamma^t r_t]$ and $J_h(\pi) := \mathbb{E}[\sum_{t \geq 0} \gamma^t c_t]$ for the reward and cost return under $\pi$, and $d_{\rho_0}^\pi$ for the discounted state-visitation distribution induced by $\pi$ from initial distribution $\rho_0$. For a reference policy $\pi'$, we define $A_r^{\pi'}(s,a) = Q_r^{\pi'}(s,a) - V_r^{\pi'}(s)$ and $A_h^{\pi'}(s,a) = Q_h^{\pi'}(s,a) - V_h^{\pi'}(s)$.

**Lemma 4** (Performance difference via TV). *For any two policies $\pi$ and $\pi'$, we have*

$$
\begin{aligned}
J_r(\pi) - J_r(\pi') &= \frac{1}{1-\gamma} \mathbb{E}_{s \sim d_{\rho_0}^\pi,\, a \sim \pi(\cdot|s)} \big[ A_r^{\pi'}(s,a) \big], \\
J_h(\pi) - J_h(\pi') &= \frac{1}{1-\gamma} \mathbb{E}_{s \sim d_{\rho_0}^\pi,\, a \sim \pi(\cdot|s)} \big[ A_h^{\pi'}(s,a) \big].
\end{aligned}
\tag{49}
$$

*Moreover, if $|r(s,a)| \leq R_{\max}$ then*

$$
\big| J_r(\pi) - J_r(\pi') \big| \leq \frac{2R_{\max}}{(1-\gamma)^2} \sup_s \mathrm{TV}\big(\pi(\cdot|s), \pi'(\cdot|s)\big),
\tag{50}
$$

*and an analogous bound holds for $J_h$ with $R_{\max}$ replaced by $C_{\max}$.*

*Proof.* The equalities are the standard performance-difference lemma obtained by unrolling the Bellman equations and telescoping the resulting series.

For the inequality, bounded rewards imply $|V_r^{\pi'}(s)| \leq R_{\max}/(1-\gamma)$ and $|Q_r^{\pi'}(s,a)| \leq R_{\max}/(1-\gamma)$ for all $(s,a)$, hence $\big| A_r^{\pi'}(s,a) \big| \leq 2R_{\max}/(1-\gamma)$. Moreover, for every $s$ we have $\mathbb{E}_{a \sim \pi'(\cdot|s)}[A_r^{\pi'}(s,a)] = 0$, so

$$
\Big| \mathbb{E}_{a \sim \pi(\cdot|s)} A_r^{\pi'}(s,a) \Big| = \Big| \mathbb{E}_{a \sim \pi(\cdot|s)} A_r^{\pi'}(s,a) - \mathbb{E}_{a \sim \pi'(\cdot|s)} A_r^{\pi'}(s,a) \Big| \leq 2 \frac{R_{\max}}{1-\gamma} \mathrm{TV}\big(\pi(\cdot|s), \pi'(\cdot|s)\big),
\tag{51}
$$

where we used the standard inequality $|\mathbb{E}_p f - \mathbb{E}_q f| \leq 2\|f\|_\infty \mathrm{TV}(p, q)$. Plugging this bound into the performance-difference lemma and taking the supremum over $s$ yields

$$\big|J_r(\pi) - J_r(\pi')\big| \leq \frac{1}{1-\gamma}\, \mathbb{E}_{s \sim d^\pi_{\rho_0}}\, qu\Big[2\frac{R_{\max}}{1-\gamma}\,\mathrm{TV}(\pi(\cdot|s), \pi'(\cdot|s))\Big] \leq \frac{2R_{\max}}{(1-\gamma)^2}\, \sup_s \mathrm{TV}\big(\pi(\cdot|s), \pi'(\cdot|s)\big). \tag{52}$$

The bound for $J_h$ follows by replacing $R_{\max}$ with $C_{\max}$. $\qquad\square$

**Proposition 2** (Policy gap under base-space KL control). *Assume that the base latent distribution $q_u(\cdot|s)$ used by the refined policy satisfies a uniform KL constraint*

$$D_{\mathrm{KL}}\big(q_u(\cdot|s)\,\|\,\mathcal{N}(\cdot)\big) \leq \varepsilon_{\mathrm{base}} \qquad \textit{for all } s, \tag{53}$$

*where $\mathcal{N}$ is the standard Gaussian base of the flow prior and we refer to the upper bound $\varepsilon_{\mathrm{base}}$ as the base-space KL radius. Let*

$$\Delta_\beta := \sup_s \mathrm{TV}\big(\pi_0(\cdot|s), \pi_\beta(\cdot|s)\big)$$

*denote the mismatch between the flow prior and the behavior policy. Then the refined policy satisfies the following reward and cost bounds:*

$$\big|J_r(\pi) - J_r(\pi_0)\big| \leq \frac{2R_{\max}}{(1-\gamma)^2}\,\sqrt{\tfrac{1}{2}\,\varepsilon_{\mathrm{base}}}, \tag{P1}$$

$$\big|J_h(\pi) - J_h(\pi_0)\big| \leq \frac{2C_{\max}}{(1-\gamma)^2}\,\sqrt{\tfrac{1}{2}\,\varepsilon_{\mathrm{base}}}, \tag{P2}$$

$$\big|J_r(\pi) - J_r(\pi_\beta)\big| \leq \frac{2R_{\max}}{(1-\gamma)^2}\,\big(\sqrt{\tfrac{1}{2}\,\varepsilon_{\mathrm{base}}} + \Delta_\beta\big), \tag{P3}$$

$$\big|J_h(\pi) - J_h(\pi_\beta)\big| \leq \frac{2C_{\max}}{(1-\gamma)^2}\,\big(\sqrt{\tfrac{1}{2}\,\varepsilon_{\mathrm{base}}} + \Delta_\beta\big). \tag{P4}$$

*Proof.* We first consider the gap between $\pi$ and the flow prior $\pi_0$. By Lemma 4 with $\pi' = \pi_0$ it suffices to control $\sup_s \mathrm{TV}(\pi(\cdot|s), \pi_0(\cdot|s))$. By Pinsker's inequality we have

$$\mathrm{TV}\big(\pi(\cdot|s), \pi_0(\cdot|s)\big) \leq \sqrt{\tfrac{1}{2}\,D_{\mathrm{KL}}\big(\pi(\cdot|s)\,\|\,\pi_0(\cdot|s)\big)}. \tag{54}$$

Lemma 3 shows that the policy KL is bounded by the base-space KL under the flow+decoder mapping:

$$D_{\mathrm{KL}}\big(\pi(\cdot|s)\,\|\,\pi_0(\cdot|s)\big) \leq D_{\mathrm{KL}}\big(q_u(\cdot|s)\,\|\,\mathcal{N}(\cdot)\big) \leq \varepsilon_{\mathrm{base}}, \tag{55}$$

and hence

$$\sup_s \mathrm{TV}\big(\pi(\cdot|s), \pi_0(\cdot|s)\big) \leq \sqrt{\tfrac{1}{2}\,\varepsilon_{\mathrm{base}}}. \tag{56}$$

Substituting this into Lemma 4 yields (P1) and (P2).

For the gap to the behavior policy, Corollary 1 in the main text implies that for each state $s$,

$$\mathrm{TV}\big(\pi(\cdot|s), \pi_\beta(\cdot|s)\big) \leq \sqrt{\tfrac{1}{2}\,D_{\mathrm{KL}}\big(q_u(\cdot|s)\,\|\,\mathcal{N}(\cdot)\big)} + \mathrm{TV}\big(\pi_0(\cdot|s), \pi_\beta(\cdot|s)\big). \tag{57}$$

Applying the uniform bounds on the base-space KL and on $\mathrm{TV}(\pi_0, \pi_\beta)$ gives

$$\sup_s \mathrm{TV}\big(\pi(\cdot|s), \pi_\beta(\cdot|s)\big) \leq \sqrt{\tfrac{1}{2}\,\varepsilon_{\mathrm{base}}} + \Delta_\beta. \tag{58}$$

Plugging this into Lemma 4 with $\pi' = \pi_\beta$ yields (P3) and (P4). $\qquad\square$

**Discussion** The above results show that FLRP admits explicit reward and cost policy-gap bounds (Proposition 2) in terms of the base-space KL radius $\varepsilon_{\text{base}}$ and th prior–behavior mismatch $\Delta_\beta$. In particular, Eqs. (P1)–(P4) make the role of $\varepsilon_{\text{base}}$ transparent: by constraining the refined base distribution $q_u(\cdot \mid s)$ to stay within a KL ball around the Gaussian base of the flow prior, we directly bound the TV/$W_2$ shift between the refined policy and the prior/behavior policies, and hence obtain tunable control over out-of-distribution (OOD) extrapolation. This aligns with the central tension in offline RL between policy improvement and staying close to the data distribution.

Within offline safe RL, LSPC (Koirala et al., 2024) also derives policy-gap and violation bounds in a CMDP setting. For example, they show that

$$V_r^{\pi^\star}(\rho_0) - V_r^\pi(\rho_0) \;\leq\; \frac{2R_{\max}}{(1-\gamma)^2}\left(\sqrt{\frac{\varepsilon_1'}{2}} + \sqrt{\frac{\varepsilon_2'}{2}}\right), \tag{59}$$

where $\pi^\star$ is the optimal safe policy and $\varepsilon_1', \varepsilon_2'$ collect approximation errors from the CVAE-based latent model and value estimators. Although similar in form, our guarantees are not a stronger version of LSPC's global policy-gap bounds, but a complementary type of result. LSPC focuses on how far the learned safe policy can be from the optimal safe policy in terms of return and constraint satisfaction, with bounds expressed via abstract approximation errors (e.g., $\varepsilon_1', \varepsilon_2'$). Our analysis instead focuses on how far refinement can move the policy away from the data/prior distribution in a flow-based latent space, and how a base-space KL regularizer keeps this shift controlled and tunable. Crucially, compared with $\varepsilon_1', \varepsilon_2'$, $\varepsilon_{\text{base}}$ is not a latent error term but a regularization parameter in the training objective: it has a direct geometric interpretation and can be monitored and adjusted in practice, providing an explicit and controllable mechanism for OOD risk suppression that is embedded into the shared refiner and Gaussian regularization.

## D IMPLEMENTATION DETAILS

In this section, we describe our experimental framework and implementation of the proposed method, including benchmark and datasets, task descriptions and evaluation metrics, and training details.

### D.1 BENCHMARK DETAILS

We use the Datasets for Safe Reinforcement Learning (DSRL) benchmark suite (Liu et al., 2023a) to train and evaluate our method as well as all baselines. DSRL provides 38 offline datasets spanning multiple safe RL environments (Safety-Gymnasium, Bullet-Safety-Gym, and Safe MetaDrive) with varying difficulty levels. These datasets follow a D4RL-style (Fu et al., 2020) API and include detailed cost signals in addition to reward returns.

For the baselines, we adopt the authors' official implementations and default hyperparameters when available (especially for FISOR and LSPC). For other methods (BCQL / BCQ-Lag, CPQ, CDT), we use the OSRL framework's implementations and settings to ensure fair comparison.

### D.2 TASK DESCRIPTIONS

Below are the three environment suites used in our experiments, with their main task types and distinguishing safety vs. complexity features. Figure 5 shows three example visualizations.

#### D.2.1 SAFETY-GYMNASIUM

Safety-Gymnasium (Ji et al., 2023) is a unified MuJoCo-based benchmark collection offering a variety of continuous control tasks (e.g. Goal, Button, Push, Circle, Velocity, etc.). Agents include Point, Car, Ant, HalfCheetah, etc. The tasks vary both in goal structure (e.g. reach a goal, push an object, navigate through buttons) and safety constraints (velocity limits, obstacle avoidance, collision cost). Some tasks include hazards or "sigwalls" that act as soft or hard boundaries. These tasks test both navigation and locomotion under safety constraints.

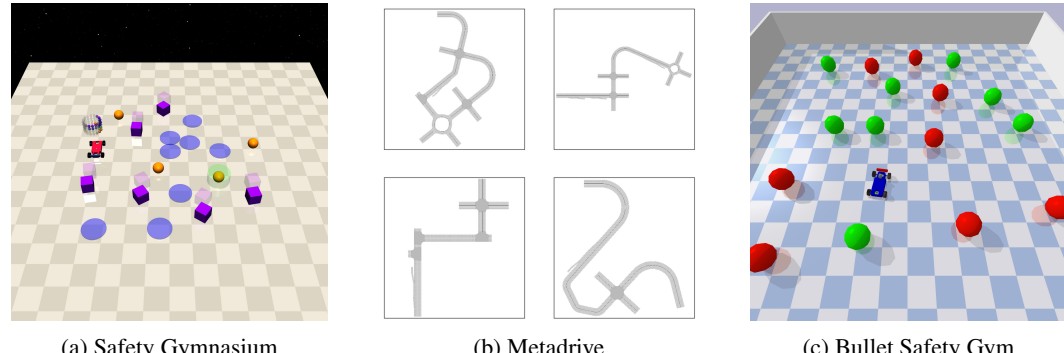

(a) Safety Gymnasium  (b) Metadrive  (c) Bullet Safety Gym

Figure 5: Example visualization from the simulation environments used in our experiments.

### D.2.2 BULLET-SAFETY-GYM

Bullet-Safety-Gym (Gronauer, 2022) is based on the PyBullet physics engine. It includes similar task types (Circle, Run, Gather, Reach) with agents such as Ball, Car, Drone, Ant. The dynamics tend to have shorter horizons and more variability in physics (collision, friction) compared to MuJoCo, which raises safety risk under state/action noise. Cost signals usually arise from collisions or from exceeding safe boundaries. This makes the tasks more challenging in terms of generalization and handling unsafe transitions.

### D.2.3 SAFE METADRIVE

MetaDrives (Li et al., 2022) is a simulator for driving/traffic/autonomous vehicle tasks under safety constraint. Its "safe RL" subset includes tasks with realistic road networks, dynamic agents, procedural map generation, traffic rules, and hazards. Observations often include vehicle state, road context; actions are continuous control of speed/steering. Safety constraints include collisions, lane infractions, and staying within road limits. These tasks are more realistic in terms of environment unpredictability, driving constraints, and possibly partial observability or environmental stochasticity.

### D.3 DATASET VISUALIZATION

We further present the distribution of offline trajectories in the cost–return space across three representative environments, as shown in Figure 6. In the `CarPush` task from Safety-Gymnasium, the reward distribution is narrow and low, while the cost spans a wide range. This results in a weak correlation between reward and safety: most trajectories incur significant costs even when achieving only modest returns, making strict constraint satisfaction particularly challenging. In the `MediumMean` task from Safe MetaDrive, the reward exhibits distinct discrete bands, each associated with a specific cost level. This reflects mode-switching behaviors and a strong reward–cost coupling; although feasible trajectories exist, achieving high reward under tight cost limits requires careful selection among these behavioral clusters. The `CarRun` task from Bullet Safety Gym demonstrates a smoother trade-off frontier, where reward gradually increases with cost, forming a continuous and diverse distribution. While safe, high-reward trajectories remain sparse, the presence of mid-reward, intermediate-cost episodes renders this dataset more amenable to constrained policy optimization compared to the other two.

### D.4 EVALUATION METRICS

We evaluate the performance of all methods using two metrics: *normalized reward return* and *normalized cost return*, following standard evaluation practices used in offline RL benchmarks like D4RL Fu et al. (2020) and adopted by recent safe RL methods such as CDT Liu et al. (2023b), LSPC (Koirala et al., 2024) and FISOR (Zheng et al., 2024). The normalized reward is defined as:

$$R_{\text{norm}} = \frac{R_\pi - r_{\min}(\mathcal{T})}{r_{\max}(\mathcal{T}) - r_{\min}(\mathcal{T})} \tag{60}$$

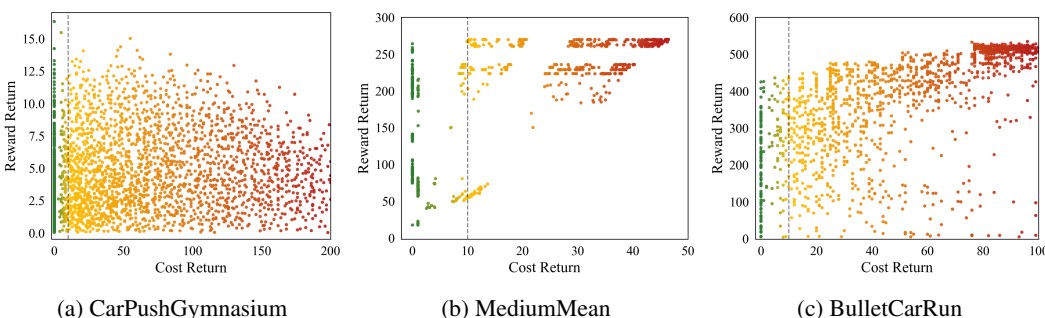

(a) CarPushGymnasium          (b) MediumMean          (c) BulletCarRun

Figure 6: Example visualization of the dataset used in our experiments.

where $R_\pi$ is the total reward return of the trained policy $\pi$, and $r_{\max}(\mathcal{T})$, $r_{\min}(\mathcal{T})$ denote the maximum and minimum reward returns observed in the dataset $\mathcal{T}$, respectively.

The normalized cost is computed as:

$$C_{\text{norm}} = \frac{C_\pi}{\kappa + \epsilon} \tag{61}$$

where $C_\pi$ is the total cost return of policy $\pi$, $\kappa$ is the cost limit, which we set to 10 for all tasks, and $\epsilon$ is a small constant added to avoid numerical instability when $\kappa = 0$.

### D.5 TRAINING DETAILS

For all baseline methods, we adopt their default hyperparameter configurations. To ensure a fair comparison across all methods, we set the rollout length for each task to match the maximum number of allowed interaction steps. The cost limit for the baselines is set to 10 for all tasks. The common key hyperparameters used for our method and baselines are shown in Table 5. Table 6 lists other key hyperparameters used for FLRP. We apply the same configuration across all tasks and environments without per-task tuning.

Table 5: Model Configuration Parameters

| Parameter | CPQ | BCQ-L | CDT | LSPC | FISOR | FLRP |
|---|---|---|---|---|---|---|
| *Common Settings:* | | | | | | |
| Training steps | | | $1 \times 10^6$ | | | |
| Batch size | | | 512 | | | |
| Discount factor | | | 0.99 | | | |
| Activate function | | | ReLu | | | |
| *Algorithm-Specific Settings:* | | | | | | |
| Hidden layer size | 256 | 256 | 256 | 256 | 256 | 256 |
| Soft update rate ($\tau$) | 0.005 | 0.005 | 0.005 | 0.005 | 0.001 | 0.001 |
| Cost limit | 10 | 10 | 10 | – | – | – |
| *Learning Rates ($\times 10^{-3}$):* | | | | | | |
| Actor learning rate | 1.0 | 1.0 | 0.1 | 0.3 | 0.3 | 0.3 |
| Critic learning rate | 1.0 | 1.0 | 0.1 | 0.3 | 0.3 | 0.3 |

The pseudocode for FLRP is provided in Algorithm 1. All experiments were conducted on eight NVIDIA RTX 6000 Ada Generation GPUs, each with 48 GB of memory. Each experiment is run with 3 random seeds, and results are averaged over 10 evaluation episodes per seed.

### D.6 COMPUTATIONAL COST

**Compute overhead of the flow prior.** As an extension of the ablation study, to quantify the computational footprint of the flow prior, we compare it against an otherwise identical refiner equipped

Table 6: Hyperparameters of FLRP.

| Parameter | Value |
|---|---|
| Expectile $\tau$ | 0.9 |
| Asymmetric L2 loss coeff | 0.9 |
| Target temperature | 3 |
| Value temperature | 5 |
| Advantage weight clip (reward) | $(-\infty,\ 100]$ |
| Advantage weight clip (cost) | $(-\infty,\ 150]$ |
| Refine steps $T$ | 3 |
| Refiner loss weight $\lambda_r, \lambda_h, \lambda_{sh}$ | 1,1,0.5 |

Table 7: Compute profile of the refiner with a Gaussian prior vs. a flow prior (identical architecture and training setup). FLOPs are per training step; NF Time fraction is the proportion of wall-clock per step spent in the prior.

| Prior | Train time / step (s) | Peak mem. (GB) | Infer. latency (ms) | Refiner FLOPs (GFLOPs/step) | Prior FLOPs (GFLOPs/call) | NF time frac. (%) |
|---|---|---|---|---|---|---|
| Gaussian | 0.052 | 1.06 | 1.21 | 0.29 | 0.00 | 0.05 |
| Flow | 0.086 | 1.07 | 2.13 | 0.48 | 0.18 | 3.33 |

with a Gaussian prior (Table 7). The flow prior increases per-step training time from 0.052,s to 0.086,s (about $\times 1.6$) and single-step inference latency from 1.21, ms to 2.13, ms (about $\times 1.7$), while peak memory remains essentially unchanged (1.06, GB vs. 1.07, GB). In terms of arithmetic cost, refiner updates require 0.29, GFLOPs per step with a Gaussian prior and 0.47, GFLOPs with a flow prior (roughly $\times 1.6$); the flow prior itself accounts for about 0.18, GFLOPs per call, corresponding to approximate 38% of the refiner's FLOPs and approximate 3.3% of the wall-clock time per training step. In contrast, the Gaussian prior baseline incurs essentially zero prior FLOPs and a negligible NF time fraction ($\approx 0.05\%$). Overall, the flow prior introduces a modest but measurable overhead, while keeping both training and inference well within a practical compute budget.

**Architectural simplicity of the flow prior.** Our normalizing flow prior is intentionally designed to be lightweight. We use a coupling-based architecture with affine transformations (RealNVP-style), so the Jacobian of each layer is triangular and the log-determinant can be computed in $O(d)$ time without any matrix inversion. Forward and inverse mappings share the same couplings and remain strictly first-order—there is no need to invert Hessians, solve inner optimization problems, or run costly fixed-point iterations. Combined with a moderate latent dimension and a small number of coupling layers, this keeps the flow prior numerically stable and computationally inexpensive while still providing exact likelihoods and invertible latent transformations.

# E  ADDITIONAL EXPERIMENTS

**Reversed expectile for feasibility function.** The reversed expectile parameter $\tau_h$ controls how conservative the feasibility critic is and thus how the safe region is learned. We sweep different $\tau_h$ values to quantify the gap between our HJ-based feasibility estimates and the true safe region constructed from the offline dataset. Intuitively, a smaller $\tau_h$ emphasizes lower $Q_h$ values, making $V_h$ more pessimistic and shrinking the induced feasible set $\{s \mid V_h(s) \leq 0\}$; this should yield high precision but low recall w.r.t. the true safe set. A larger $\tau_h$ has the opposite effect, expanding the feasible set and trading precision for recall. Table 8 confirms this trend on both CarRun and AntCircle: as $\tau_h$ increases, recall consistently improves while precision decreases. The effect is much sharper on AntCircle, whose safe region is more complex, indicating that harder tasks require a more optimistic critic (larger $\tau_h$) to achieve comparable coverage of the true safe set.

---

**Algorithm 1** FLRP Training (Two-Stage)

---

**Require:** Offline dataset $\mathcal{D}$
1: Init critics $(Q_r, V_r)$, $(Q_h, V_h)$; flows $p_\phi, q_\psi$; decoder $\pi_\theta$; refiners $\{R_s, R_r, R_{\text{sh}}\}$

2: **Stage 1: Critic and flow pretraining**
3: **while** not converged$_{\text{base}}$ **do**
4:     Sample minibatch $(s, a, r, c, s') \sim \mathcal{D}$, draw $z_q \sim q_\psi(z \mid s, a)$

5:     **// Critic updates**
6:     Update safety critics $(Q_h, V_h)$ by HJ-style backup               ▷ Eq. 8, Eq. 9
7:     Update reward/value critics $(Q_r, V_r)$ by TD / advantage targets      ▷ Eq. 21, Eq. 22

8:     **// Flow prior and decoder**
9:     Compute BC loss and prior-shaping loss $\mathcal{L}_{\text{flow}}$              ▷ Eq. 12
10:    Update $p_\phi, q_\psi$ and $\pi_\theta$ using flow objective               ▷ Eq. 13
11: **end while**

12: **Stage 2: Latent refiner training (freeze base model)**
13: Freeze parameters of $(Q_r, V_r)$, $(Q_h, V_h)$, $p_\phi, q_\psi$, and $\pi_\theta$
14: **while** not converged$_{\text{ref}}$ **do**
15:    Sample minibatch $(s, a, r, c, s') \sim \mathcal{D}$, draw $z_q \sim q_\psi(z \mid s, a)$
16:    Compute base code $u_q \leftarrow T_\phi^{-1}(z_q \mid s)$

17:    *(a) Safety refiner*
18:    $u_s \leftarrow R_s(u_q, s)$
19:    Decode $\tilde{a}_s \leftarrow \pi_\theta\big(T_\phi(u_s \mid s), s\big)$
20:    Compute safety loss $\mathcal{L}_h$ from $Q_h(s, \tilde{a}_s)$                ▷ Eq. 14
21:    Update parameters of $R_s$ w.r.t. $\mathcal{L}_h$

22:    *(b) Reward refiner*
23:    $u_r \leftarrow R_r(u_q, s)$
24:    Decode $\tilde{a}_r \leftarrow \pi_\theta\big(T_\phi(u_r \mid s), s\big)$
25:    Compute reward loss $\mathcal{L}_r$ from $Q_r(s, \tilde{a}_r)$ (masked by feasibility)    ▷ Eq. 15
26:    Update parameters of $R_r$ w.r.t. $\mathcal{L}_r$

27:    *(c) Shared refiner / OOD control*
28:    $u_{\text{sh}} \leftarrow R_{\text{sh}}(u_q, s)$
29:    Compute base-space regularizer $\mathcal{L}_{\text{sh}}$ (e.g., KL to $\mathcal{N}(0, I)$ or $\|u_{\text{sh}}\|^2$)    ▷ Eq. 16
30:    Update parameters of $R_{\text{sh}}$ w.r.t. $\mathcal{L}_{\text{sh}}$
31: **end while**

---

Table 8: Sensitivity of the HJ-based feasibility classifier to the expectile parameter $\tau_h$. Recall/precision are computed on the offline buffer on task by treating steps from zero-cost trajectories as ground-truth safe.

| Task | Metric | $\tau_h$ | | | | |
|------|--------|------|------|------|------|------|
| | | 0.6 | 0.7 | 0.8 | 0.9 | 0.95 |
| CarRun | Recall | 0.32 | 0.39 | 0.54 | 0.76 | 0.85 |
| | Precision | 0.76 | 0.68 | 0.51 | 0.24 | 0.21 |
| AntCircle | Recall | 0.04 | 0.08 | 0.27 | 0.79 | 0.88 |
| | Precision | 0.78 | 0.42 | 0.06 | 0.05 | 0.05 |

**Decoder freezing ablation.** Freezing the decoder is a core modeling choice in our method: the theoretical coupling between the latent prior and the refiner—and the resulting bounds on action and policy shift—critically rely on the decoder remaining fixed. Allowing the decoder to change

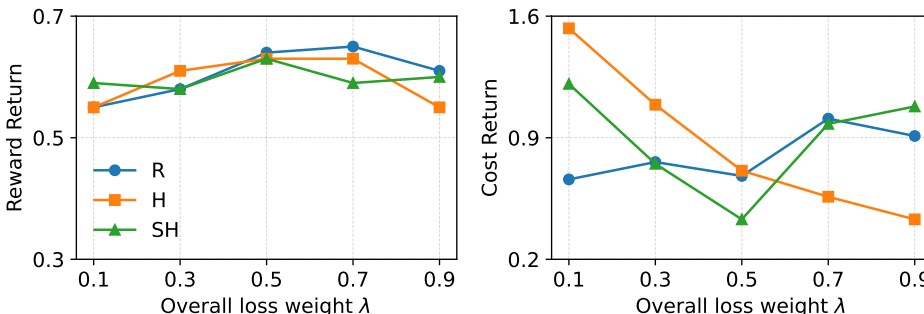

Figure 7: Effect of refiner loss weights on FLRP performance: varying the relative weights of the reward (R), safety (H), and shared (SH) refiners yields a robust response and enables a smooth trade-off between reward return and cost.

would break this coupling and make both the analysis and the interpretation of the refinement steps much less clear. To quantify how much performance is potentially sacrificed by this restriction, we compare our default "frozen decoder" training with an alternative scheme where the refiner and decoder are updated in alternating phases. The result is shown in Table 9a. On the simpler task `CarRun`, the two variants achieve very similar performance: with a frozen decoder, we obtain a reward of 0.87 at zero cost, while alternating updates yield a reward of 0.84, also at zero cost. On the more challenging `AntCircle` task, alternating updates increase the reward from 0.45 to 0.69, but at the price of a higher cost (from 0.25 to 0.56). Thus, while partially unfreezing the decoder can improve returns on complex tasks, it does so by relaxing safety, whereas the frozen-decoder variant preserves our theoretical guarantees and achieves tighter cost control.

**Effect of refiner loss weights.** Figure 7 investigates how the relative weights assigned to the three refiners (R, H, SH) affect performance. Overall, FLRP is quite robust: within a broad range of loss weights, the reward and cost curves remain stable without sudden degradation. When the safety refiner H is severely under-weighted (left part of the curves), the policy becomes noticeably less safe, confirming that H is the main driver toward low-cost regions. As the weight of H increases, the policy consistently moves to safer operating points. In contrast, putting more emphasis on the reward refiner R tends to increase the reward return, but also leads to higher cost, which is consistent with its role of exploiting high-return directions near the constraint boundary. The shared refiner SH behaves like a regularizer: when its weight is too small, the policy becomes less coordinated and slightly more unstable; when its weight is too large, over-regularization harms both reward and safety. The best performance is obtained

Table 9: Decoder and refiner ablations.

(a) Frozen vs. alternating decoder

| Task | Reward | Cost |
|---|---|---|
| CarRun (frozen) | 0.87 | 0.00 |
| CarRun (alter.) | 0.84 | 0.00 |
| AntCircle (frozen) | 0.45 | 0.25 |
| AntCircle (alter.) | 0.69 | 0.56 |

(b) Refiner optimization strategy on `AntCircle`

| Refiner Deisgn | Reward | Cost |
|---|---|---|
| Decoupled 3-refiners | 0.45 | 0.25 |
| Single unified refiner | 0.07 | 0.00 |
| Averaged 3-refiners | 0.51 | 0.45 |

for intermediate SH weights, where it can effectively absorb residual interactions between R and H while keeping the refinement close to the flow prior. These trends show that (i) FLRP's performance is not overly sensitive to the exact choice of refiner weights, and (ii) by tuning the relative weights of R, H, and SH, practitioners can smoothly control the reward–cost trade-off without changing the underlying critics or flow model.

**Ablation on refiner optimization strategy.** We further investigate whether the three-refiner architecture is really necessary, or whether one can obtain similar behavior by changing only the optimization scheme while keeping the same total loss. On `AntCircle`, we fix the loss weights $(\lambda_r, \lambda_h, \lambda_{sh})$ and compare our default design—three decoupled refiners (H, R, SH) optimized sequentially—with two alternatives (Table 9b): (i) a single unified refiner, which directly optimizes

the sum of the three refiner losses, and (ii) an averaged 3-refiner update, where we still learn three refiners but average their latent updates before applying a single step to the base code.

The results show that the three-refiner design is crucial for obtaining a good reward–cost trade-off. The unified refiner collapses to an overly conservative solution (reward 0.07, cost 0.00): because a single set of parameters must simultaneously satisfy safety, reward, and regularization objectives, the gradients from these components frequently conflict, and the optimizer converges to a compromise that prioritizes low cost but fails to exploit high-return directions. By contrast, the averaged-update variant achieves high reward (0.51) but with much higher cost (0.45): averaging the three latent updates at a single point mixes conflicting safety and reward gradients, partially canceling the safety correction and diluting the shared refiner's regularization, which leads to high-return but unsafe solutions. Our sequential H→R→SH updates (0.45 reward, 0.25 cost) strike a substantially better balance that cannot be mimicked by a single averaged step. Overall, these results indicate that separating safety, reward, and shared refiners—each with its own parameters and update direction—is more effective than collapsing them into a single refiner or naively averaging their gradients.

