# OpenReview forum: "Flow-Guided Latent Refiner Policies for Safe Offline Reinforcement Learning"
_ICLR.cc/2026/Conference — Submitted to ICLR 2026_

### Official Review · Reviewer_ZJrt · 2025-10-27

**Soundness:** 1
**Presentation:** 3
**Contribution:** 1
**Rating:** 2
**Confidence:** 4

**Summary:**

This paper studies the problem of ensuring hard constraints and avoiding out-of-distribution in safe offline reinforcement learning. The proposed method first models the latent action through a flow module, which concentrate probability mass on safe regions, and then applies a refinement stage that updates the base-space variable to improve reward and safety before decoding actions. Experimental results on several benchmarks show that the proposed method achieves fewer violations and comparable or higher return than the baselines.

**Strengths:**

1. The experiments are extensive, covering mainstream safe offline RL benchmarks and algorithms.
2. The writing of the paper is clear.

**Weaknesses:**

1. The contribution of the proposed method over existing methods is unclear.
2. The motivation of the refinement stage given the flow module is not described clearly.
3. The practice of the refinement steps is not sound enough.

**Questions:**

1. This paper considers a state-wise hard constraint setting. The authors claim that existing methods for handling hard constraints typically induce conservatism. What is the cause of such conservatism?
2. The authors claim that the proposed method is not "pushed by hard constraints" but "pulled by density". What does this mean? What is the benefit of being "pulled by density" over being "pushed by hard constraints"?
3. In prior density shaping, the loss function (12) involves the value functions with regard to reward. Does this mean that the flow module is also taking reward into consideration? If this is the case, it seems that both the flow module and the refiners are simultaneously considering reward and constraint. Then, what is the difference between the roles of them?
4. In the refinement stage, why are the three refiners applied in the order of safety-reward-shared? Why not averaging their incremental updates into a single update or even learning a single refiner that directly optimizes the total loss? Does the process of repeating the refinement steps finally converge?

---

> ### Author Response · Authors · 2025-11-21
> **Response to the reviewer ZJrt's comments (Part one)**
>
> **W1: Contribution over existing methods is unclear**
>
> We appreciate the reviewer’s concern. Conceptually, FLRP combines three ingredients—HJ-based feasibility, a conditional flow prior, and latent refiners—but the goal is not a pure engineering blend. Compared with prior flow-based offline RL such as Let Offline RL Flow (CNF), our work targets the safe offline RL setting with hard/near-zero-violation constraints, and leverages the invertibility of flows and a Gaussian base distribution to derive explicit base-space KL bounds that propagate to TV/$W_2$ bounds on action-distribution shift and OOD mass (Corollary 1). This property is not available for CVAE- or diffusion-based approaches, which lack exact likelihood and a tractable change-of-variables structure, nor is it the focus of LSPC, whose theory provides global performance/constraint gaps with respect to an optimal safe policy rather than geometric control of OOD. Algorithmically, FLRP instantiates these guarantees via a density-shaped flow prior and a small-step refiner protocol that couples safety, reward, and OOD control under a frozen decoder.
>
> We have revised the introduction and related work, and added an extended discussion in Appendix B that contrasts FLRP with other safe RL methods, and highlights these new properties.
>
> **W2: Motivation of the refinement stage & W3: The practice of the refinement steps**
>
> Our intention is to use the flow module and the refiners in complementary roles rather than as redundant components. The conditional flow and encoder are trained (with Eq. (12)) to learn a latent action manifold that places higher density on empirically safe and high-return behavior, and to provide an explicit Gaussian base space where KL-based bounds on distribution shift and OOD mass are tractable. On its own, this produces a conservative, behavior-like prior policy but does not perform iterative policy improvement. The refinement stage is then responsible for injecting the critic information into this prior in a controlled way: the safety refiner moves latent codes toward HJ-feasible regions, the reward refiner improves reward within that feasibility-guided manifold, and the shared refiner regularizes the updates back toward the Gaussian base to keep the induced policy within the provable OOD bounds. In other words, the flow module builds a safety-aware latent substrate with geometric guarantees, while the refiners perform policy optimization on top of it.
>
> As for the practice of three-refinement design, theoretically, the guarantees in Lemmas 2–3 and Corollary 1 depend only on the overall refinement map being non-expansive in base space and on the resulting base-space KL remaining bounded; they are agnostic to the specific ordering of safety/reward/shared refiners or to whether the losses are implemented via one or multiple modules. Our choice of a sequential Safety \(\rightarrow\) Reward \(\rightarrow\) Shared update is mainly motivated by optimization and interpretability. If we collapse all objectives into a single refiner with a single loss, the gradients from safety, reward, and KL/OOD regularization often point in conflicting directions, which leads to noisy updates that are harder to tune and to reason about. Splitting the refinement into three small steps allows us to first pull latents toward the HJ-feasible region (safety), then improve return within this feasibility-guided manifold (reward), and finally apply a Gaussian/KL step to control OOD mass and keep actions on support (shared).
>
> We have provided ablation studies in the main text (Figure 3) comparing the performance of the no-refinement variant and alternative refiner orders. The results show that all refiner variants significantly outperform the no-refine baseline, confirming that latent refinement is the key driver of performance gains. Additionally, we have included a new ablation in Appendix E, Table 9, demonstrating that the decoupled three-refiner setup effectively mitigates gradient conflicts and yields a better safety–reward trade-off compared to single-step or entangled updates.

---

> ### Author Response · Authors · 2025-11-21
> **Response to the reviewer ZJrt's comments (Part two)**
>
> **Q1: The conservatism of hard constraint setting**
>
> In the state-wise hard-constraint setting we consider, a policy must satisfy $h(s_t) \leq 0$ at every timestep along the trajectory. This strict requirement, when combined with offline uncertainty, naturally leads to conservative policy behavior. We elaborate below on several contributing factors:
>
> - **Smaller feasible region during training.**
>   The hard constraint excludes any action or trajectory that violates the safety condition at any timestep, which effectively shrinks the feasible set $\mathcal{A}_{\text{safe}}(s)$ from which the policy can choose. This limits exploration and optimization, especially in high-dimensional tasks where the safe region may be narrow or nonconvex.
>
> - **Limited coverage of safe data in offline datasets.**
>   In realistic datasets, especially in safety-critical tasks (e.g., autonomous driving, locomotion), strictly safe trajectories may constitute only a small fraction of the dataset. Most offline data include near-safe or slightly unsafe behaviors. Since hard-constraint methods must discard these, they are forced to learn from a smaller and potentially suboptimal subset of the data.
>
> - **No tolerance for trade-offs.**
>   Unlike soft-constrained or density-based methods that allow small violations in favor of higher reward, hard-constraint methods treat all violations equally disqualifying. This removes any possibility of controlled risk-reward balancing and encourages the policy to stay deeply within the certified safe set.
>
> - **Safety propagation compounds conservatism.**
>   Since the constraint must hold at every timestep, safety propagation across long horizons becomes multiplicative: even one uncertain or unsafe step invalidates an entire trajectory. This creates a bottleneck during learning and encourages overly cautious strategies.
>
> Existing methods reflect these issues: CPO [1] uses worst-case cost estimates to optimize within an approximate safe set, pushing the policy inward from constraint boundaries. Offline methods such as FISOR [2] and LSPC [3] explicitly avoid uncertified regions: FISOR constructs a feasible region based on offline support and optimizes only within it; LSPC builds a conservatively safe latent space before allowing any reward improvement in the latent space. While these methods ensure zero-violation safety, they do so at the cost of overly conservative behavior.
>
>
> Our method is still conservative but in a different, more structured way. We use the feasibility critic to shape a flow prior so that high base-space density is assigned only to actions that are both feasible and high-reward, while actions that are clearly unsafe or unsupported by data receive low density. The refiner then operates in this base space with a violation penalty on $Q_h(s, \bar{a})-V_h(s)$ and proximal regularization, so it naturally avoids regions that are simultaneously low-density and predicted unsafe, but does not automatically reject all boundary-adjacent states: as long as the critic certifies feasibility and the flow assigns sufficient density, the policy can approach the constraint boundary without being pushed back solely due to lack of coverage. This is the sense in which we differ from existing state-wise hard-constraint methods that rely on a pessimistically shrunk hard safe set.
>
> **References**
>
> [1] Achiam et al., *Constrained Policy Optimization*, ICML 2017.
> [2] Zheng et al., *Safe Offline Reinforcement Learning with Feasibility-Guided Diffusion Model*, ICLR 2024.
> [3] Koirala, et al. *Latent Safety-Constrained Policy Approach for Safe Offline Reinforcement Learning*, ICLR, 2025.

---

> > ### Author Response · Authors · 2025-11-21
> > **Response to the reviewer ZJrt's comments (Part three)**
> >
> > **Q2: “Pushed by hard constraints” vs. “pulled by density”**
> >
> > By “pushed by hard constraints,” we refer to methods that first construct an explicit feasible set and then enforce safety by projecting policy updates back into this set. For example, CPO and projection-based CMDP methods perform a reward-improving step and then solve a constrained update or projection that keeps the policy inside a hard safe region [1,2]. Similarly, RCRL and FISOR use HJ-style feasibility to identify a feasible region and train a policy to stay inside this region [3,4]. In all these cases, the safe set is estimated offline and typically chosen pessimistically; any update that tries to cross its boundary is “pushed back” by a hard constraint, which can lead to conservative behavior—especially when the feasible set is shrunk due to uncertainty.
> >
> > In contrast, by “pulled by density” we mean that safety and in-distribution behavior are encoded in the shape of a latent density, and the policy is regularized to move toward high-density regions rather than being projected onto a hard set. Concretely, we learn a conditional flow prior $p_\phi(z \mid s)$ and decoder $\pi_\theta(a \mid s, z)$ using a feasibility-weighted ELBO: samples that are both well-supported by the data and certified feasible by the HJ-based critics receive higher weights, so their latent codes concentrate in high-density regions. During refinement, we operate in the base Gaussian and perform small updates that improve reward and safety while being regularized by the flow/decoder density; moving into low-density regions increases the loss, so gradients “pull” the refined action back toward safety-dense, on-support regions. This is analogous in spirit to prior offline RL work that uses generative models or likelihood terms to keep policies near the data support [5,6], but here the density is explicitly shaped by feasibility.
> >
> > The benefit of this density-based mechanism is fourfold. First, it is more robust to approximation error in the cost/HJ critic. Hard-constraint methods require an explicit feasible-set boundary, so small local errors in $Q_h$ or $V_h$ can shrink this boundary and exclude genuinely safe actions. In contrast, our method does not depend on an exact feasibility boundary: critic imperfections only perturb the density and violation terms smoothly, rather than abruptly changing whether an entire region is admissible. Second, it provides a smooth, differentiable notion of safety and data support: instead of a non-smooth projection onto a pessimistically shrunk safe set, refinement follows continuous gradients of the density and critics, allowing the policy to approach constraint boundaries whenever the critic certifies feasibility and the flow assigns sufficient density. Third, within the feasibility-shaped high-density manifold, the refiner can still move toward high-reward directions near the constraint boundary as long as the critic indicates safety, so FLRP can exploit boundary-adjacent, high-return actions that hard-set projection methods routinely discard, reducing unnecessary conservatism while maintaining near-zero violations. Fourth, the approach offers built-in OOD control: high flow/decoder density corresponds to behavior-supported, feasibility-weighted regions, so being “pulled by density’’ naturally keeps the refined policy in-distribution and in empirically safe regions, without requiring separate KL or trust-region penalties to avoid distribution shift.
> >
> > We have added a one-sentence description in Section 3.2 to clarify the meaning of the two different methods.
> >
> > **References**
> >
> > [1] Achiam et al., *Constrained Policy Optimization*, ICML 2017.
> > [2] Yang et al., *Projection-based Constrained Policy Optimization*, 2020.
> > [3] Yu et al., *Reachability-based Safe Reinforcement Learning*, 2022.
> > [4] Zheng et al., *Safe Offline Reinforcement Learning with Feasibility-Guided Diffusion Model*, ICLR 2024.
> > [5] Fujimoto et al., *Off-Policy Deep RL without Exploration*, ICML 2019.
> > [6] Kumar et al., *Stabilizing Off-Policy Q-Learning via Bootstrapping Error Reduction (BEAR)*, NeurIPS 2019.

---

> ### Author Response · Authors · 2025-11-21
> **Response to the reviewer ZJrt's comments (Part four)**
>
> **Q3: Difference between the roles of the flow module and the refiners**
>
> We agree that both components make use of reward and safety information, but they play distinct and complementary roles.
>
> The flow module is trained to perform global, geometry-level shaping of the action manifold. It uses the reward and feasibility critics only through sample weights in a generative modeling objective, assigning higher base-space density to actions that are both well supported by the data and certified feasible. This yields a smooth, safety-aware latent manifold and provides an OOD shield, but by itself, it does not perform any per-state policy improvement.
>
> In contrast, the refiners are lightweight policies that operate on the Gaussian base latent during policy optimization and evaluation. Given a state, they apply a small number of residual updates to the base variable, guided directly by actor–critic style losses that depend on $Q_r$ and $Q_h$ (e.g., via safety and reward advantages), together with proximal and density regularization. Their role is to carry out local policy improvement within the safety-shaped manifold defined by the flow.
>
> In other words, the flow prior defines *where* it is safe and data-supported to search (a global, feasibility-shaped latent geometry), while the refiners decide *how* to move within this geometry for each state to trade off reward and safety.
>
> We have added an ablation in the main text on the order of refiners, which contains no refiner schedule. This means we directly use the sampled action from the flow. As shown in Figure 3, all refiner variants significantly outperform "No refine", confirming that latent refinement is the key driver of gains.
>
> **Q4: Why this refiner order and design?**
>
> In the refinement stage, we apply three refiners in the fixed order Safety $\rightarrow$ Reward $\rightarrow$ Shared. This choice is deliberate rather than arbitrary. The safety refiner is applied first to pull the base latent toward critic-certified feasible regions by reducing the safety advantage $Q_h(s,\bar a)-V_h(s)$. The reward refiner is then applied on top of this safer latent to exploit high-return directions while remaining within the safety-shaped manifold. Finally, the shared refiner acts as a residual coordination step: it absorbs remaining interactions between reward and safety and incorporates density/proximal regularization to keep the update close to the flow prior. In our theoretical analysis we abstract the composition of refiners as a single refiner map, so the order does not affect the distribution-shift bounds; the safety–reward–shared ordering is a practical choice for optimization stability and interpretability.
>
> We also considered alternatives such as averaging the three incremental updates into a single update or learning a single refiner that directly optimizes the total loss. In both cases, all gradients (reward, safety, density) are mixed into one update, which makes training more sensitive to the relative weighting between objectives and less stable when the gradients conflict (e.g., reward vs. safety near the constraint boundary). To support this claim, we have added two ablations. First, we vary the refiner order on representative tasks on four agents, Car, Ant, Ball, and Drone (Main text, Figure 3): the model is reasonably robust to swapping safety and reward (H$\rightarrow$R$\rightarrow$SH vs. R$\rightarrow$H$\rightarrow$SH), while randomizing the order increases variance. This suggests that our method does not hinge on a brittle ordering, but placing the shared refiner last is important to consistently regularize and coordinate the preceding updates. Second (Appendix E, Table 9), we fix the refiner losses and compare our default design—three decoupled refiners (H, R, SH) optimized sequentially—with (i) a single unified refiner and (ii) an averaged 3-refiner update. The unified refiner collapses to an overly conservative solution (very low cost but almost no reward), while the averaged 3-refiner update attains a high reward at the price of much higher cost. In contrast, the sequential H$\to$R$\to$SH scheme achieves a significantly better reward–cost balance.
>
> Regarding convergence, the refinement is implemented as a shallow residual network of fixed depth $T$, not as an unrolled iterative optimization scheme. We do not iterate to a fixed point and therefore do not rely on classical convergence guarantees; instead, we treat $T$ as an architectural hyperparameter. Moreover, the refiner losses include proximal and density terms that prevent large deviations from the base latent, which stabilizes inference. As shown in Figure 4, an ablation over the number of refinement steps indicates that performance saturates after a small number of steps, suggesting that a few refinement layers are sufficient in practice.

---

> ### Comment · Reviewer_ZJrt · 2025-11-27
>
> Thanks for the rebuttal. My concerns regarding the role and soundness of the refinement steps remain unresolved. The authors claim the flow module provides a "global, geometry-level shaping of the action manifold" without any "per-state policy improvement". However, the output of the flow is conditioned on the state, thus it is a already per-state policy instead of a uniform one. The necessity of further "policy improvement" is unclear. Is there any reason that the flow and decoder cannot learn a good enough policy? Furthermore, is there any guarantee that the refinement steps produce a policy that is safer and achieves higher rewards over the original flow and decoder? It seems that the theories in the paper only say "if the refinement is not too large, the policy will not deviate too much". But staying close to the original policy does not necessitates an additional refinement.

---

> ### Author Response · Authors · 2025-11-27
>
> Thank you for your reply. We would like to clarify a possible misunderstanding. We do not claim that the flow policy is independent of state. The flow and decoder together define a state-conditioned action distribution, i.e., a per-state policy. In fact, a non–state-conditioned flow would be unusable in RL, as it would produce the same action distribution regardless of state, which is clearly infeasible for decision-making.
>
> Our point is more subtle: although this distribution is per-state, it is not trained to directly optimize reward or constraint satisfaction in a per-state, greedy manner. Rather, the flow is fitted via density estimation on behavior data, combined with feasibility- and reward-based density shaping (Eq. 12), instead of standard policy-gradient or actor–critic updates. In this sense, the flow provides a globally trained, geometry-level shaping of a feasible and data-supported action manifold that is biased toward high-return, feasible actions, but is not expected to select reward-optimal actions for each state. Therefore, the flow and decoder model a “safe” latent manifold of likely feasible behaviors, shaped by behavior, feasibility, and soft reward signals—but without direct per-state reward optimization, **they still do not guarantee high-return actions.**
>
> The refinement step is then introduced to fulfill this role by performing local, critic-guided improvements in return (while preserving safety and staying in-distribution). To clarify this empirically, we have already added a “No refine” ablation across four tasks (Car, Ant, Ball, Drone) in Figure 3. These results show that all refinement schedules significantly outperform “No refine” in return, at the cost of slightly higher violations—demonstrating that refinement indeed brings improvement. Aside from this empirical evidence, the theoretical bounds in Appendix C.8 help justify our design, but they do not imply that refinement is unnecessary.
>
> This reflects a common optimization paradigm also seen in LSPC and related latent-policy methods: one first constructs or identifies a feasible and high-potential latent manifold, and then performs task-specific optimization within this region. In particular, LSPC constructs a safety-aware latent space using a CVAE; but experiment shows simply sampling from this prior does not yield high-performing policies, so an additional latent encoder is trained to guide optimization. Our design follows a similar motivation but more explicitly decouples this process through three refiners.

---

### Official Review · Reviewer_89xo · 2025-10-27

**Soundness:** 3
**Presentation:** 3
**Contribution:** 2
**Rating:** 4
**Confidence:** 4

**Summary:**

The paper addresses the problem of safe offline reinforcement learning (RL), proposing a new method termed FLRP. The key idea is to leverage a normalizing-flow-based latent action manifold together with a refiner module that operates in the latent space to balance safety and performance while remaining within the support of the offline dataset. The method is evaluated on standard safe offline RL benchmarks and demonstrates competitive results against recent baselines such as FISOR, LSPC, and CDT.

While the empirical results are promising and the paper is well-organized, the conceptual novelty appears somewhat limited given the overlap with several recent works. Addressing the following points (please refer to Weaknesses and Questions sections) could significantly strengthen the paper’s contributions.

**Strengths:**

1. The formulation of a zero-violation feasibility critic via HJ-style value functions is a natural and well-motivated choice for safety-critical domains. The latent-space refinement idea is also practical and nicely complements this design.
2. The framework avoids the complex minimax optimization often arising from Lagrangian formulations in constrained RL, leading to a more tractable training procedure.
3. The paper is generally well-written and clearly structured, and it includes ablation studies that help justify the design choices.

**Weaknesses:**

1. Conceptual novelty and prior work: The incorporation of HJ-based critics for safety has already been explored in works such as RCRL [1] and FISOR [2], while decoding from safety-dense latent embeddings is reminiscent of LSPC[3]. Although these works are cited, the specific novelty and distinction of FLRP are not sufficiently articulated. Strengthening the introduction section and related-work discussion to clearly delineate what is new (and what is adapted) would improve the paper’s contribution clarity.
2. Potential conservatism: The use of feasibility values might make the method conservative when high-reward and low-cost regions do not coincide with high-density behavior data. This could explain why the authors primarily report results under a fixed cost threshold (10) rather than varying thresholds (e.g., 10, 20, 40) as done in the original DSRL evaluation. Including results under different safety budgets would help evaluate the robustness–safety trade-off.
3. Computational complexity: Normalizing flow modeling in high-dimensional action spaces can be computationally expensive due to inverse computations and potential convergence sensitivity. While implementation details are briefly discussed, it would be interesting to know whether implicit density models (e.g., flow-matching or diffusion-based methods) could serve as lighter or more stable alternatives.
4. Presentation and algorithm clarity: Algorithm 1 is currently deferred to the very last of the paper. Given that the proposed method involves multiple components (flow model, multiple refiners, and multiple critics), including a compact pseudocode or schematic in the main text (or at least referencing the appendix early) would significantly aid readability and reproducibility.
5. Ablation on HJ-feasibility function: Although the reward-wise policy improvement scheme remains the same, the “w/o HJ” variant consistently underperforms (reward-wise) the full FLRP model. Clarifying whether the critics or refiners are retrained independently for this variant would help ensure that this comparison is fair and interpretable.
6. Figure 2 clarity: While the visualization itself is appealing and informative, the labels are not clearly defined and the figure is not thoroughly discussed in Section 5. Clearer explanations of what each axis and color corresponds to would help readers better connect the figure to the text.

[1] Yu, et al. "Reachability constrained reinforcement learning." ICML, 2022.

[2] Zheng, et al. "Safe offline reinforcement learning with feasibility-guided diffusion model." ICLR, 2024.

[3] Koirala, et al. "Latent Safety-Constrained Policy Approach for Safe Offline Reinforcement Learning." ICLR, 2025

**Questions:**

1. Both methods FISOR and FLRP use an HJ-based feasibility critic, yet FLRP appears *generally* less conservative (achieving higher reward at a cost of higher cost). Could the authors clarify what aspect of FLRP’s design makes it less conservative than FISOR?
2. LSPC derives policy-gap and performance bounds similar to those in Corollary 1 of this paper. This is not discussed in the paper. How do the theoretical guarantees of FLRP differ or improve upon those results?
3. How sensitive is FLRP’s performance to the order of refiners? Would reversing or randomizing the order affect stability or outcomes?
4. Figure 2 interpretation:
* The authors claim a zero-violation scheme. If the full action space were visualized, would some regions correspond to zero Q_h values? If only the cost refiner were used, would it drive the samples toward that region?
* It also seems that maximizing the decoder log-density alone already steers actions toward safer regions. Is this effect specific to the CarRun environment or more general across tasks?
5.  Equation (14): Could the authors elaborate on the safety-expert AWR objective? The form looks unusual, and I could not find a directly similar objective in the cited references.

---

> ### Author Response · Authors · 2025-11-21
> **Response to the reviewer 89xo's comments (Part one)**
>
> **W1: Conceptual novelty and prior work.**
>
> We thank the reviewer for pointing this out. We agree that our method builds on two existing lines of work: (i) HJ-style feasibility critics for safety, as in RCRL[1] and FISOR[2], and (ii) latent action manifolds and decoders, as in PLAS[3]/LSPC[4]. Our contribution is not to re-introduce these components, but to couple them in a new way via a flow-based latent prior and refiner protocol that yields explicit and controllable guarantees on distribution shift. Concretely, unlike FISOR, FLRP uses an invertible conditional flow with exact log-likelihood and analytic Jacobian, allowing us to monitor a base-space KL and propagate it into TV/$W_2$ bounds, and to drive latent refiners under a frozen decoder so that safety and OOD control are tightly linked. Unlike LSPC and related latent-action methods (which typically use VAE-style embeddings and heuristic conservatism), FLRP uses the flow density itself as a regularizer, together with HJ-based feasibility, to implement feasibility-aware, in-distribution refinement with theoretical control of OOD mass. To the best of our knowledge, no prior work combines (a) a flow-based latent policy prior with exact likelihoods, (b) formal base-space OOD bounds that are directly tied to this prior, and (c) a multi-stage refiner that couples safety, reward, and OOD control under a frozen decoder. This combination leads to a new, tunable notion of conservatism for safe offline RL that is not present in earlier HJ-based or latent-action methods.
>
> To clarify these distinctions, we have revised the introduction and related works section, and added an extended section in Appendix B.1. The introduction now explicitly separates components adapted from prior work from our novel contributions (flow-based density control, the refinement protocol, and OOD performance bounds). In the related works, we additionally introduce a new discussion of HJ-based safe RL methods and clarify how FLRP differs from prior uses of HJ reachability. The extended section reviews existing generative latent-space offline RL methods—including PLAS,  LSPC, and others—and then presents a side-by-side table (Table 4) comparing FLRP to these approaches in terms of modeling choices, safety coupling, and theoretical guarantees.
>
> **References**
>
> [1]  Yu, et al. *Reachability constrained reinforcement learning*, ICML, 2022.
> [2]  Zheng, et al. *Safe offline reinforcement learning with feasibility-guided diffusion model*, ICLR, 2024.
> [3]  Zhou, et al. *Plas: Latent action space for offline reinforcement learning*, CoRL, 2021.
> [4]  Koirala, et al. *Latent Safety-Constrained Policy Approach for Safe Offline Reinforcement Learning*, ICLR, 2025.

---

> ### Author Response · Authors · 2025-11-21
> **Response to the reviewer 89xo's comments (Part two)**
>
> **W2: Potential conservatism and evaluation under different safety levels**
>
> We appreciate the concern that feasibility-based regularization, combined with density modeling, might make FLRP overly conservative—especially when high-return, low-cost regions do not perfectly align with high-density behavior data. Our method, however, follows the same hard/near-zero-violation paradigm as HJ-based approaches like FISOR: the primary objective is to suppress violations as much as possible, rather than to treat cost as a tunable budget. This reflects safety-critical applications (e.g., robotics in shared spaces, grid control), where even small violations may be unacceptable. In this sense, our formulation is conceptually different from budgeted safe RL methods (e.g., CDT[5], CPQ[6]), in which a cost budget is injected as an explicit design knob; we will clarify this distinction more clearly in the revised paper.
>
> Importantly, FLRP is not simply “following” the raw behavior density. During prior training, we use the shaping loss in Eq.(12) to reorganize the flow prior so that empirically feasible, high-quality actions become high-density in the Gaussian base space. Concretely, Eq.~(12) upweights samples with large reward advantage $Q_r(s,a) - V_r(s)$ that are also marked feasible by the HJ critic, and pulls their latent codes $u = T_{\phi}^{-1}(z_q \mid s)$ toward the high-density region of the base distribution. As a result, the learned prior concentrates mass on “high-return and low-cost” regions whenever they are supported by the dataset, rather than simply mirroring the most frequent behavior.
>
> Finally, to directly probe the return–cost trade-off, we conducted a hyperparameter ablation where we vary the relative loss weights of the three refiners (reward, safety, and shared; i.e., the $\lambda$s in Eq.(17)) and report the resulting average return and cost. The new curves (Appendix E, Figure 7) show that FLRP is robust over a broad range of refiner weights: only extreme under-weighting of the safety refiner leads to noticeably higher cost. As the weight of the safety refiner increases, the policy consistently moves toward safer solutions (lower cost) with only moderate return degradation, whereas emphasizing the reward refiner yields a higher return at the expense of higher cost. The shared refiner behaves like a regularizer, with both very small and very large weights hurting performance, and the best return–cost trade-off obtained at intermediate values. These trends indicate that, even within a near-zero-violation regime, FLRP does not collapse to an overly conservative solution and provides a smooth, interpretable knob to balance performance and safety by tuning the refiner loss weights.
>
> [5] Liu, et al. *Constrained decision transformer for offline safe reinforcement learnin*, ICML, 2023
> [6] Xu, et al. *Constraints penalized q-learning for safe offline reinforcement learning*, AAAI, 2022

---

> > ### Author Response · Authors · 2025-11-21
> > **Response to the reviewer 89xo's comments (Part three)**
> >
> > **W3: Computational complexity of normalizing flows and possible alternatives**
> >
> > We agree that flow-based models can, in principle, be expensive if used naively in high-dimensional spaces. In FLRP, however, we deliberately use a lightweight conditional flow with affine coupling layers and triangular Jacobians, so that both the forward and inverse mappings and the log-determinant are inexpensive to compute. The actual architecture used in our experiments is very small (three coupling blocks with shallow MLPs), and we will make this explicit in the implementation details section to emphasize that no costly Hessian- or matrix-inversion operations are involved.
> >
> > To make the overhead concrete, we report detailed compute statistics in Appendix D.6 (Table 7), comparing our coupling–flow prior to a Gaussian prior with the same refiner and decoder. The flow prior increases training time only moderately (0.052 s to 0.086 s per step, approx. 65\%) and inference latency from 1.21 ms to 2.13 ms, with virtually unchanged peak memory (1.06 GB vs. 1.07 GB); refiner FLOPs grow from 0.29 to 0.48 GFLOPs/step and each prior call costs 0.18 GFLOPs, yet the normalizing flow still accounts for only about 3.3\% of total training time. Overall, this overhead is modest relative to the consistent return–safety gains over the Gaussian prior reported in Table 3. We have also added a discussion in Appendix D.6 on the architectural simplicity of our RealNVP-style flow prior, emphasizing its lightweight coupling layers with $O(d)$ Jacobian log-determinants and exact, invertible mappings without any inner optimization.
> >
> > Regarding alternatives such as flow-matching or diffusion-based models: these are indeed attractive implicit density estimators, but they do not provide the same explicit change-of-variables structure and exact likelihoods as normalizing flows. Our theoretical results crucially rely on the flow’s bijectivity and analytic Jacobian to (a) monitor a base-space KL and (b) propagate it into TV/$W_2$ bounds on distributional shift, which in turn are used to regularize the latent refiners under a frozen decoder. Replacing the flow with an implicit model would thus require a substantially different analysis and is orthogonal to the main contribution of this paper.
> >
> > **W4: Presentation and algorithm clarity**
> >
> > We thank the reviewer for this suggestion. In the revision, we have expanded Algorithm 1 to explicitly decompose each iteration into critic updates, flow/decoder updates, and three separate refiner steps (safety, reward, shared), and we indicate which equations (HJ backup, flow objective, refiner objective) are used at each line. And we have added a more compact description of training in the main text section 3.4 with a clear reference to Algorithm 1. This makes the training loop and the role of each refiner clearer and improves reproducibility.

---

> > > ### Author Response · Authors · 2025-11-21
> > > **Response to the reviewer 89xo's comments (Part four)**
> > >
> > > **W5: Ablation on HJ feasibility function**
> > >
> > > We thank the reviewer for asking for more details on the “w/o HJ” variant. In this ablation, we do not remove feasibility guidance altogether; instead, we replace the HJ-based feasibility signal with a purely cost-based heuristic, like the implementation of the LSPC and FISRO ablation study. Concretely, we train a standard cost value function and declare a state–action pair feasible if its predicted cost lies below the empirical 75th percentile of the zero-violation samples in the dataset. This yields a binary feasibility indicator based only on cost statistics, which then substitutes the HJ signal everywhere it is used in FLRP: in prior density shaping (Eq.12) and in the masks used by the refiners. The flow prior, decoder, critics, and refiners are all retrained from scratch under this modified feasibility definition, with identical architectures and training budgets; the only change is the source of the feasibility signal (HJ reachability vs. cost thresholding).
> > >
> > > Under this configuration, it is expected—and consistent with FISOR’s observations—that the “w/o HJ” variant can be both more conservative and less effective in terms of return. The cost-based thresholding produces noisier and inflated infeasible regions: many genuinely safe, high-return actions are misclassified as infeasible, so the prior and refiners are trained on a much smaller and biased subset of the behavior data. This over-conservatism hurts final returns, even though the procedure is ostensibly “reward-driven.” In contrast, HJ reachability propagates safety constraints through the dynamics and yields a more structured and robust notion of feasibility, which leads to lower violations and higher returns in our experiments.
> > >
> > > We have revised the ablation on the HJ-feasibility part so that this comparison is fully transparent and interpretable.
> > >
> > > **W6: Figure 2 interpretation**
> > >
> > > We appreciate the reviewer’s comment. Our intention with Figure 2 is to illustrate, on a fixed CarRun state, how the three refiners (safety, reward, shared) pull the latent action in different directions and how the shared refiner keeps updates on support. We agree that the current presentation could be clearer for readers seeing the figure for the first time.
> > >
> > > In the revision, we have (i) expanded the caption to state explicitly that each panel shows a 2D slice of the action space (x-axis: velocity, y-axis: steering angle), with colors corresponding to $Q_h(s,a)$, $Q_r(s,a)$, and decoder log-density $\log \pi_\theta(a\mid s)$ in panels (a), (b), and (c), respectively, and (ii) added a few sentences in Section~5 walking through panels (a)–(c), explaining how the reward and safety refiners move toward distinct high-value regions while the shared refiner balances them and stays in the high-density area of the decoder.

---

> > > > ### Author Response · Authors · 2025-11-21
> > > > **Response to the reviewer 89xo's comments (Part five)**
> > > >
> > > > **Q1: Relative conservatism of FISOR vs. FLRP**
> > > >
> > > > Thank you for raising this point. We would first like to clarify the empirical picture in Table 1. When averaged over tasks in each benchmark, FLRP does not trade safety for performance: on Safety-Gymnasium, FLRP achieves higher return and lower cost than FISOR (avg. reward 0.33 vs. 0.29, cost 0.18 vs. 0.40); on Bullet-Safety-Gym, the same pattern holds (avg. reward 0.54 vs. 0.43, cost 0.04 vs. 0.17). On Safe MetaDrive, FLRP attains substantially lower cost (0.19 vs. 0.38) at slightly lower return (0.34 vs. 0.40), i.e., is actually more conservative overall. Thus, FLRP is not generally “less conservative” in the sense of violating constraints more often; in most settings, it improves return while keeping costs equal or lower than FISOR.
> > > >
> > > > That said, on some individual tasks, FLRP does obtain a higher return at a slightly higher cost, and we agree it is useful to understand why this can happen despite both methods using an HJ-based feasibility critic. Conceptually, FISOR uses the HJ signal in a very conservative way: a diffusion policy generates multiple candidate actions, and the final action is chosen by re-ranking them according to predicted safety, effectively picking the safest sample in each state. This tends to bias the policy toward the interior of the safe set, sacrificing boundary high-return regions. In contrast, FLRP uses HJ reachability as a soft guidance signal: (i) during prior density shaping, we increase density in regions that are both high-return and feasible, and (ii) during refinement, the safety refiner penalizes violations while the reward and shared refiners optimize return within the high-density, HJ-feasible manifold controlled by the flow. As a result, FLRP can exploit high-return actions near the safety boundary while remaining in-distribution and below the prescribed violation thresholds.
> > > >
> > > > **Q2: Relation to LSPC’s policy-gap and performance bounds**
> > > >
> > > > We thank the reviewer for pointing out this connection. LSPC indeed derives performance and constraint-violation bounds for its latent-space safe offline RL algorithm. At a high level, those results compare the learned policy to an optimal safe policy in a CMDP setting: under assumptions on approximation error and on the discrepancy between the CVAE-induced safe policy and the optimal safe policy, LSPC bounds the value gap and the constraint violation gap, and further provides a sample-complexity rate. These bounds are formulated in terms of policy/distribution discrepancies and function-approximation errors, not in terms of a particular generative model’s geometry.
> > > >
> > > > By contrast, Corollary~1 in our paper answers a different but complementary question. We work in a flow+decoder setting and study how latent refiners change the induced action distribution when we regularize in the Gaussian base space. Leveraging the invertibility and exact likelihood of conditional flows, we show that controlling a base-space KL divergence directly controls the downstream distributional shift in action space, in TV/$W_2 $metrics, and hence the OOD mass with respect to the data-supported region. This is a geometric, change-of-variables-based–based bound that explicitly ties refinement updates to the flow prior, and it is specific to the flow-based construction used in FLRP.
> > > >
> > > > Thus, our guarantees are not a stronger version of LSPC’s global policy-gap bounds, but a complementary type of result. LSPC focuses on how far the learned policy can be from the optimal safe policy in terms of return and constraint satisfaction, under CVAE-based latent modeling. Our analysis instead focuses on how far refinement can move the policy away from the data/prior distribution in a flow-based latent space, and how a base-space KL regularizer keeps this shift controlled and tunable. This latter property is what we exploit algorithmically via the shared refiner and Gaussian regularization: the bound is directly embedded into the training objective, providing an explicit and controllable mechanism for OOD risk suppression.

---

> > > > > ### Author Response · Authors · 2025-11-21
> > > > > **Response to the reviewer 89xo's comments (Part six)**
> > > > >
> > > > > **Q3: Sensitivity of FLRP to the order of refiners**
> > > > >
> > > > > We thank the reviewer for raising this question. The guarantees in Section 4 (Lemmas 2–3 and Corollary 1) are order-agnostic: the bounds on TV/$W_2$ distance and OOD mass depend on the overall refinement map and its base-space KL, not on the specific sequencing of the safety, reward, and shared refiners. We now clarify this explicitly in Appendix B.6.
> > > > >
> > > > > To empirically probe this, we have added an ablation in the main text (Figure 3) comparing four schedules on CarRun and AntCircle: Safety$\rightarrow$Reward$\rightarrow$Shared, Reward$\rightarrow$Safety$\rightarrow$Shared, a random permutation per update, and a “No refine’’ baseline. All refiner variants significantly outperform No refine, confirming that latent refinement is the key driver of gains. On CarRun, the two fixed orders achieve similar high returns with near-zero cost, indicating limited sensitivity. On AntCircle, they trace a smooth return–cost trade-off (safety-first is slightly safer; reward-first slightly higher return), while the random schedule lies in between but with higher variance. These trends suggest that FLRP is reasonably robust to refiner order, and that our chosen Safety$\rightarrow$Reward$\rightarrow$Shared schedule with the shared refiner last is a stable and interpretable default.
> > > > >
> > > > > **Q4: Interpretation of Figure 2 and the role of decoder density**
> > > > >
> > > > > **1. Zero-violation region and cost refiner only.** Our zero-violation guarantee is expressed through the feasibility critic $Q_h$: by construction, $Q_h(s,a)\le 0$ certifies the existence of a policy whose entire trajectory from $(s,a)$ satisfies the constraint (Definition~1 and Eqs. (5)--(7)). In practice, we approximate $Q_h$ via the feasible Bellman operator and reversed-expectile regression, so the learned critic forms a band of low values around the zero level set rather than a perfectly flat region at exactly zero. Figure 2 visualizes a 2D slice of this critic for one particular state in CarRun, hence small local fluctuations around zero are expected.
> > > > >
> > > > > We have added an ablation on the total loss weights of each refiner (Appendix E, Figure 7). When the safety refiner dominates the loss (i.e., set $\lambda_h=0.9$ and $\lambda_r=\lambda_{sh}=0.05$), the updates are driven almost purely by the safety advantage and move samples toward low-$Q_h$ regions, as reflected by decreased normalized cost. This comes at the expense of lower reward, confirming that a “cost-only’’ refiner is overly conservative and motivating the full combination of safety, reward, and shared refiners.
> > > > >
> > > > > **2. Decoder log-density versus safety**  The decoder/flow is not trained on the raw behavior distribution but on a *feasibility-weighted* distribution:
> > > > >
> > > > > $$
> > > > > \tilde{p}_D(s,a) \propto w(s,a)\,p_D(s,a),
> > > > > $$
> > > > >
> > > > > with $w(s,a) = \sigma(-Q_h/T_q)\,\sigma(-V_h/T_v)$ (Eq.(11)). Thus, a high decoder log-likelihood indicates actions that are both on-support and assigned a relatively large feasibility weight. This induces a positive correlation between log-density and safety, but does *not* make density a sufficient certificate of safety:
> > > > >
> > > > > - The weights $w(s,a)$ are smooth rather than hard indicators, so mildly unsafe or boundary actions still receive non-zero weight and are repeatedly seen during training.
> > > > > - In realistic datasets, non-strictly-safe behaviors remain a large fraction of the data, especially in tasks where high reward and low cost conflict. Even after reweighting, frequently occurring but unsafe behaviors can still occupy relatively high-density regions.
> > > > >
> > > > > For this reason, we treat decoder log-density mainly as an on-support proxy, while the zero-violation property is enforced by combining the HJ-based critic, the prior-shaping loss (which explicitly pulls feasible high-reward actions toward high base-space density), and the safety refiner.
> > > > >
> > > > > In the CarRun example of Figure 2, the environment is relatively simple and the dataset contains a high proportion of safe trajectories, so maximizing decoder log-density happens to move samples toward safer regions. In more complex tasks, the high-density and low-$Q_h$ regions can be noticeably misaligned.

---

> ### Author Response · Authors · 2025-11-21
> **Response to the reviewer 89xo's comments (Part seven)**
>
> **Q5. Safety expert loss**
>
> We thank the reviewer for pointing out that Eq.(14) looks unusual compared to standard AWR and that the objective was not fully specified. There is indeed a typesetting error in the current version: one term in the safety expert loss was accidentally dropped. As stated in the main text, the intended loss for the safety expert $f_h$ has a push--pull form and is *not* a single penalty on $Q_h$, but a combination of a violation term and a safety-weighted behavior regression term:
>
> $$
> L_h = \mathbb{E}_{(s,a)\sim\mathcal D} \left[
>     \phi\big(Q_h(s,\bar a(s,u_T)) - V_h(s)\big)
>     + w_h(s,a)\ \|\bar a(s,u_T) - a\|_2
> \right],
> $$
>
> where $\bar a(s,u_T)$ is the decoded refined action after $T$ refinement steps, $\phi(\cdot)$ is a soft penalty (e.g., softplus), and
>
> $$
> w_h(s,a) = \exp\left(-{Q_h(s,a) - V_h(s)}/{\beta_h}\right) \cdot \mathbf{I}_{\text{feas}}(s,a)
> $$
>
> is a safety-based weight. We have corrected Eq.(14) in the main text accordingly.
>
> ---
>
> **Intuition: push–pull structure**
>
> This loss decomposes into a *push* and a *pull* component:
>
> - The first term, $\phi(Q_h(s,\bar a) - V_h(s))$, directly penalizes the positive safety advantage $A_h(s,\bar a) = Q_h(s,\bar a) - V_h(s)$ of the *refined* action. Its gradient flows through $Q_h$ and the decoder to push $\bar a(s,u_T)$ back toward the feasible region $Q_h \leq 0$ whenever a violation is predicted.
>
> - The second term is a safety-weighted behavior regression: it pulls $\bar a(s,u_T)$ toward dataset actions $a$ with weights
>   $w_h(s,a) \propto \exp(-A_h(s,a)/\beta_h)$, so safer actions (lower safety advantage) provide stronger supervision.
>
> This structure mirrors AWR (which uses $\mathbb{E}[w(s,a)\,\ell(\pi(a \mid s))]$ with $w(s,a) \propto \exp(A(s,a)/\beta)$),
> but we replace reward advantage with *safety* advantage, and the log-policy loss $-\log \pi(a \mid s)$ with a deterministic BC loss $\|\bar a - a\|_2$.  The weighting mechanism is directly analogous.

---

> ### Comment · Reviewer_89xo · 2025-11-22
>
> Thank you for the detailed rebuttal response. My initial review contained many technical questions and concerns, and I appreciate the effort the authors put into addressing them.
>
> The updated pseudocode (alg. 1) is significantly clearer, and referencing it in the main text improves readability. The additional ablation studies are helpful, and the revised interpretation of Figure 2 is much improved. Thank you also for correcting the equation I flagged; please double-check the remaining equations as well, given the number of components and subscripts involved in the method, it is easy to miss.
>
> Overall, I found the discussion constructive and I am leaning toward acceptance. However, my concerns about the degree of novelty still remain. As another reviewer noted, the work can feel more like a comprehensive engineering synthesis than a fundamentally new contribution; I share this view to some extent. That said, the paper provides an interesting and thorough empirical study, and with appropriate attribution the contribution is still meaningful. To help strengthen the clarity and positioning of the work, I offer the following suggestions:
>
> 1. Regarding the newly added Table 4: If page limits permit, please consider moving it into the main text. If space is constrained, including only the safe RL baselines would still be helpful. Additionally, please include FISOR as a baseline, given its conceptual similarity to the proposed framework. This will further help delineate the contributions and design choices of your method.
>
> 2. Please clarify the comparison of the policy gap and performance bounds with related work. Due to space limitations, including this in the appendix with a pointer from the main text is acceptable. If time permits, deriving at least some mathematical expressions would help address novelty concerns raised by multiple reviewers, including myself. Otherwise please extend and include the discussion (Q2) that you already have written in your response above.
>
> 3. Suggestion: Since “flow” can refer to both normalizing flows and flow-matching, adding a footnote (e.g., early in Section 2) clarifying that “flow” denotes normalizing flows throughout would reduce potential confusion.

---

> ### Author Response · Authors · 2025-11-23
> **Response to Reviewer 89xo**
>
> We sincerely thank the reviewer for the constructive feedback and for the continued discussion, which have greatly improved the paper. We have carefully revised the manuscript according to your suggestions.
>
> ---
>
> **(1) Table 4**
>
> As suggested, we have moved Table 4 into the main text (Related Work) and added FISOR as an additional baseline. We also expanded the surrounding discussion to clarify how FLRP relates to prior generative policy methods and why it is advantageous.
>
> ---
>
> **(2) Policy gap / performance bounds and comparison with prior work**
>
> Following your suggestion, we have:
>
> - Derived explicit reward and cost policy gap / performance bounds bounds (Proposition 2 and Eqs. (P1)–(P4) in Appendix C.8), which relate the performance and constraint gaps between the refined policy and the flow prior / behavior policy to:
>   - the **uniform upper bound on the base-space KL** $\varepsilon_{\text{base}}$, and
>   - the **prior–behavior mismatch** $\Delta_\beta
> := \sup_s \mathrm{TV}\bigl(\pi_0(\cdot|s),\pi_\beta(\cdot|s)\bigr)$.
> - Added an explicit pointer in the main text (end of Section 3.3) to the new analysis in **Appendix C.8**.
>
> We then contrast these guarantees with LSPC's global performance/constrained-optimality bounds: LSPC focuses on how far the learned safe policy can be from the optimal safe policy (in terms of return and cost, via abstract approximation errors such as $(\varepsilon_1,\varepsilon_2)$, whereas our analysis focuses on how far refinement can move the policy away from the prior distribution in a flow-based latent space, and how the base-space KL regularizer provides a geometric and tunable handle on OOD shift. Compared with $\varepsilon_1,\varepsilon_2$,  $\varepsilon_{\text{base}}$  is not a latent error term but a regularization parameter in the training objective. We also integrated the core points from our earlier Q2 response directly into this appendix section, as you suggested.
>
> ---
>
> **(3) Clarifying the meaning of “flow”**
>
> We have added a footnote early in the Section 2 to avoid potential confusion between normalizing flows and flow-matching.
>
> ---
>
> We hope these revisions adequately address your concerns

---

> ### Comment · Reviewer_89xo · 2025-11-28
>
> Thanks for the additional clarification. Overall, it was a great rebuttal discussion. I am updating the score to 6 to reflect my current assessment of the work.

---

### Official Review · Reviewer_sYAh · 2025-10-31

**Soundness:** 3
**Presentation:** 3
**Contribution:** 3
**Rating:** 6
**Confidence:** 3

**Summary:**

This paper proposes FLRP (Flow-guided Latent Refiner Policies), a safe offline RL framework that addresses hard safety constraints and out-of-distribution (OOD) action issues. The method has three main components: (1) **Feasibility-based value functions** using Hamilton-Jacobi (HJ) reachability to identify safe states/actions from offline data, (2) **Conditional normalizing flows** that model the latent action manifold and concentrate probability mass on empirically safe, high-density regions, and (3) **Expert refiners** (safety, reward, shared) that perform small residual updates in the base Gaussian latent space to jointly optimize reward and safety. The framework operates entirely offline and provides theoretical guarantees that refinement in base space controls distributional shift. Experiments across 26 tasks from Safety-Gymnasium, Bullet-Safety-Gym, and Safe MetaDrive show violation rates of 0.18 vs. 0.40 (second-best baseline) while maintaining competitive returns.

**Strengths:**

- Creative combination of normalizing flows with latent-space refinement and HJ-based feasibility signals provides a principled approach to safe offline RL
- Lemmas 2-3 and Corollary 1 provide formal guarantees that controlling base-space KL bounds downstream distributional shift in Wasserstein and TV metrics
- Avoids brittle Lagrangian penalty tuning by shaping density via flows and performing feasibility-aware refinement

**Weaknesses:**

- HJ operator with sparse cost signals can undervalue genuinely safe but rare samples, introducing bias (acknowledged in Section 7)
- Requires tuning expert loss weights (λr, λh, λsh), temperatures (Tv, Tq), prior shaping coefficients, and expectile τ—though authors use single config across tasks

**Questions:**

1.  Table 3 shows flow prior helps, but what is the computational cost? Could a simpler multimodal prior (e.g., mixture of Gaussians) achieve similar benefits?

2.  Can you quantify the gap between HJ-based feasibility estimates and true safe regions? How often do safe actions get incorrectly labeled as infeasible?

3. Freezing the decoder constrains optimization to the learned manifold. How much performance is lost vs. fine-tuning the decoder? Could you alternate freezing/unfreezing?

4.  Both use feasibility guidance with diffusion/flow models. What are the key differences? When would FISOR be preferred over FLRP?

---

> ### Author Response · Authors · 2025-11-21
> **Response to the reviewer sYAh's comments (Part one)**
>
> **Q1: Computational cost of the flow model**
>
> Thank you for raising this point. In the revision, we report detailed compute statistics in Appendix D.6, Table 7, comparing our coupling–flow prior against a simple Gaussian prior with the same refiner and decoder. The flow prior increases training time from 0.052,s to 0.086,s per step (approx. 65\%) and inference latency from 1.21, ms to 2.13, ms, while peak memory remains virtually unchanged (1.06, GB vs.\ 1.07, GB). Refiner FLOPs grow moderately (0.29→0.48 GFLOPs/step) and each prior call costs 0.18 GFLOPs, yet the normalizing flow accounts for only 3.33\% of the total training time. Overall, the additional overhead of the flow prior is modest relative to the consistent return–safety improvements over the Gaussian prior reported in Table 3.
>
> Regarding simpler multimodal priors such as mixtures of Gaussians (MoG): our design is motivated by the theoretical guarantees we derive. The flow prior is bijective and provides exact log-likelihoods with analytic Jacobians, which we use to (a) monitor the KL divergence in the base space and propagate it into TV/W$_2$ bounds on distributional shift, and (b) anchor latent-space refinements to the density via a frozen decoder. Mixture-of-Gaussians priors lack these properties: they are non-invertible and do not support change-of-variables analysis, making them incompatible with our theoretical framework. This is also why we selected a Gaussian prior (not MoG) as our baseline comparison, as it preserves analytical tractability and serves as a natural contrast. Exploring lower-complexity multimodal priors (e.g., MoG) within this framework is an interesting direction, and we will mention this as future work, though it lies outside the core focus of this paper.
>
> **Q2: Gap between HJ-based feasibility estimates and true safety regions**
>
> We thank the reviewer for raising this insightful question. To quantify the conservatism of the HJ-based feasibility signal, we performed a detailed offline evaluation across a range of expectile coefficients $\tau \in \{0.6, 0.7, 0.8, 0.9, 0.95\}$, which control the conservativeness of the learned feasibility critic $V_h^{\tau_h}(s)$ (see Appendix E, Table 8). For each $\tau$, we trained a separate critic, used it to classify offline data as feasible if $V_h^\tau(s) \leq 0$, and compared these predictions against ground-truth safety labels obtained from zero-cost trajectories. We then report recall (fraction of truly safe states classified as feasible) and precision (fraction of states in the learned feasible set that are truly safe) to quantify the gap between HJ-based feasibility estimates and the true safe region.
>
> The results confirm the intended behavior of the reversed expectile. Smaller $\tau_h$ makes the critic more conservative: the reversed expectile loss emphasizes lower $Q_h$ values, biases $V_h$ toward pessimistic estimates, and shrinks the feasible set, yielding high precision but lower recall (tight safe set with some safe states rejected as infeasible). Larger $\tau_h$ has the opposite effect: the critic becomes more optimistic, the feasible set expands, recall increases, and precision decreases (higher coverage but more unsafe states leaking into the feasible region). This trade-off is smooth on CarRun, while AntCircle, which has a more complex safe set, exhibits a sharper transition—very small $\tau_h$ almost collapses the feasible region, and a more optimistic critic is required to recover high coverage. Following FISOR’s practice, we set $\tau_h = 0.9$ in our main experiments, which achieves a recall of approximately 80\% across both tasks. Therefore, around 20\% of truly safe states are incorrectly classified as infeasible, quantifying the degree of conservatism induced by our feasibility critic. This represents a reasonably conservative yet inclusive approximation of the safe region, balancing generalization with safety.

---

> ### Author Response · Authors · 2025-11-21
> **Response to the reviewer sYAh's comments (Part two)**
>
> **Q3: Decoder freezing trade-off**
>
> We appreciate this question. Freezing the decoder is crucial to our design: the theoretical coupling between the flow prior $ p_\phi(z \mid s) $, the refiner updates in base space, and our KL/TV bounds on action and policy shift all assume a fixed decoder. If the decoder is updated during refinement, this coupling breaks and the interpretation of the refiner as operating on a stable latent manifold becomes much less clear.
>
> In practice, “fine-tuning the decoder” during refinement is equivalent to alternating between refiner and decoder updates, since in both cases the decoder is optimized with the refinement objective. To quantify the resulting trade-off, we compared our default frozen-decoder setup with an alternating scheme where, after flow pretraining, we periodically unfreeze the decoder and update it together with the refiner. On the simpler CarRun task, the two variants behave similarly: the frozen version achieves a reward of 0.87 at zero cost, while the alternating version yields  0.84 reward, also at zero cost. On the more challenging AntCircle task, alternating updates increase reward from 0.45 to 0.69, but at the price of higher constraint violations (cost increases from 0.25 to 0.56).
>
> Thus, partially unfreezing the decoder can improve returns on complex tasks, but it does so by relaxing safety and giving up the clean theoretical guarantees that rely on a fixed decoder. We have added the ablation table and discussion to Appendix E, Table 9.
>
> **Q4: Differences between FISOR and FLRP**
>
> We appreciate this question. Both FLRP and FISOR employ a feasibility-guided method to enforce safety, but differ significantly in modeling choices, theoretical properties, and flexibility.
>
> 1. *Modeling:* FISOR uses a diffusion-based model for policy generation, while FLRP adopts a flow-based latent prior to learn the latent action distribution. The flow model enables exact likelihood computation, invertible mapping, and closed-form change-of-variables, which we leverage to derive base-space KL bounds on distribution shift (TV/$W_2$) and explicitly regularize the policy to remain in-distribution. In contrast, FISOR's diffusion process is sampling-based and less amenable to explicit density modeling.
>
> 2. *Optimization:* Compared with FISOR, FLRP decouples the decoder and performs lightweight latent-space refinement steps, allowing modular updates while preserving a fixed decoder. This design supports fast inference and tractable regularization.
>
> 3. *Control and tunability:* FLRP supports explicit control of conservatism via density shaping and KL regularization, enabling trade-offs between return and constraint satisfaction. FISOR is often strongly conservative by construction, but does not provide a principled way to adjust this trade-off.
>
> 4. *Use cases:* FISOR may be preferred in scenarios demanding strict conservatism and where inference cost is less critical. FLRP is well-suited for applications requiring flexible, tunable trade-offs and theoretical guarantees on distribution shift, with more efficient refinement during deployment.

---

> > ### Comment · Reviewer_sYAh · 2025-11-24
> > **Follow up discuss**
> >
> > Thanks for your detailed response. However, I still have some concerns.
> >
> > FISOR uses IQL and a KL regularizer to mitigate OOD issues. You state that “FISOR is often strongly conservative by construction, but does not provide a principled way to adjust this trade-off.” However, in your main experiment in Table 1, your method attains a lower average reward than FISOR while also having a lower cost. Given that the normalized cost limit is 1 and both methods’ costs are well below this threshold (your cost is 0.19 and FISOR’s is 0.38), this seems to indicate that FISOR is actually performing better in terms of the reward–cost trade-off, rather than your method.
> >
> > Moreover, you argue that the use cases for your method and FISOR are different, but the experiment you provide is exactly the same setup as in the original FISOR paper. From the current presentation, it is not clear to me where this difference in use cases manifests.
> >
> > As a result, your response has, if anything, raised additional concerns for me. I believe that more evidence and analysis are needed to clarify these points. If you can provide such evidence convincingly, I would be willing to reconsider and raise my score to 6.

---

> ### Author Response · Authors · 2025-11-24
> **Response to Reviewer sYAh**
>
> We thank the reviewer again for the careful reading and for raising these follow-up questions.
>
> At the outset, we would like to clarify that, although FISOR is formulated with a KL-ball constraint
> $D_{\mathrm{KL}}(\pi \,\|\, \pi_\beta) \le \varepsilon$ at the theoretical level, this constraint is handled analytically and absorbed into the exponential reweighting of the behavior policy which is fixed. In the official implementation, FISOR therefore
> does not employ an explicit KL regularizer.
> Instead, OOD mitigation is achieved implicitly via advantage-weighted behavior cloning/diffusion within the HJ-based feasible set. We also included this in the newly added Table 4.
>
> **The first concern**
>
> We fully acknowledge that FISOR is able to balance reward and safety in practice. Calling FISOR “strongly conservative” is therefore not meant as a criticism, but as a description of its design philosophy. We would also like to emphasize that FLRP itself is designed from a conservative viewpoint as well. FISOR builds an HJ-based feasible set and then pre-specifies different weights for different feasibility levels when training the diffusion policy. Once this feasible set and the associated weighting scheme are chosen, the resulting policy is naturally quite conservative, which is what we mean “by construction”, and there is no simple, explicit knob whose sole role is to smoothly tune the reward–safety trade-off. In particular, the method does not expose something analogous to our refiner weights or base-space KL regularizer that one can directly adjust to move along the frontier.
>
> By contrast, our method is deliberately designed to make this trade-off flexible. Because the refiners are decoupled, their order and loss weights act as explicit control knobs. In the new ablation studies we added (Fig. 3 and Fig. 7), we show that: (i) when we put the reward refiner first, on AntCircle both reward and cost increase compared to the safety-first order; and (ii) when we change the training weights on the reward and safety losses, increasing the reward (resp. safety) weight systematically pushes the solution toward higher reward and higher cost (resp. lower cost and lower reward). These experiments are precisely intended to demonstrate that FLRP can realize a range of operating points on top of the same flow prior by adjusting a small set of interpretable hyperparameters.
>
> Regarding **Table 1**, our goal there is not to claim that FISOR has a better reward–cost trade-off. Rather, we show that FLRP offers both stronger performance and lower cost on several benchmarks. On **Safety-Gymnasium**, our method achieves 0.33 reward vs. 0.29 for FISOR, with 0.18 vs. 0.40 cost; on **Bullet-Safety-Gym**, it attains 0.54 vs. 0.43 reward, with 0.04 vs. 0.17 cost. Only on **Safe MetaDrive** does FISOR achieve a higher reward (0.40 vs. 0.34 for FLRP), but at the cost of substantially more violations (0.38 vs. 0.19 for FLRP). Thus, the sentence “your method attains a lower average reward than FISOR while also having a lower cost” therefore describes the **MetaDrive** results specifically, but not Table 1 as a whole.
>
> We did not intend to over-emphasize this difference in the text, but rather to present the empirical results as they are. If any sentence in the current wording could be interpreted as a stronger claim than this, we are happy to soften it. We would like to note that the points under discussion (e.g., the trade-off statement and the following “use case” phrasing) were raised only in the rebuttal thread and are not claims made in our manuscript. Our main point is to highlight that FLRP provides a more transparent and tunable mechanism for navigating reward–safety trade-offs on top of a conservative foundation, rather than to make a definitive statement about which method has the “better” trade-off behavior overall.
>
> **The second concern**
>
> Regarding the second point, we agree that our earlier response about “different use cases” may have been ambiguous. It was intended as a general remark about deployment-oriented considerations in real-world practice, not as a claim that our method and FISOR use different benchmarks or experimental setups, which is also beyond the core scope of our paper.
>
> In the experiments, we naturally adopt well-established, widely used environments to evaluate and compare different methods. To ensure a fair, apples-to-apples comparison, we follow the original FISOR setup and use the same DSRL benchmarks. In fact, using the original configuration is necessary to faithfully reproduce the performance reported in the FISOR paper, and only then can we make a meaningful and fair comparison. This kind of “reusing the same benchmark and configuration” is standard practice in the offline RL community, and using a different evaluation protocol would only obscure the source of performance differences.
>
> ---
>
> We hope these clarifications adequately address your concerns

---

> > ### Comment · Reviewer_sYAh · 2025-11-24
> > **Follow up discuss on 1. Trade-off; 2. Use Cases**
> >
> > Thank you for the detailed response. I have two further suggestions. (1) For Table 1, you could add an overall average reward and cost to more clearly summarize the performance. (2) For the use cases, although you present results in Fig. 3, they only cover two tasks, so it is unclear whether the same trend holds more generally. Since there is still time during the rebuttal period, you might consider adding results for 2–3 additional tasks in Fig. 3 to strengthen the claim. With these improvements, I will raise my score to 6.

---

> ### Author Response · Authors · 2025-11-25
> **Response to Reviewer sYAh**
>
> Thank you for your suggestion. While we appreciate the idea of reporting a single average across all tasks in Table 1, we respectfully believe that such an aggregation, even with normalization, might be misleading and less informative, as different benchmark families still vary substantially in task difficulty, reward structure, and episodic variance. For this reason, we chose to follow prior work such as LSPC and FISOR, which do *not* report a single global average across all tasks. Instead, we have now updated Table 1 to rename the “Average” row (now labeled as Safety-Gym Avg, Bullet-SG Avg, and MetaDrive Avg) to provide a clearer and more interpretable summary, while still allowing side-by-side comparison across methods. That said, we do not believe this formatting choice affects the core findings or insights of our paper.
>
> For your second suggestion on the refiner-order ablation, we have extended the experiment from two to four tasks, covering four agents: Car, Ball, Ant, and Drone. The updated Figure 3 shows that the same qualitative trend persists across these tasks: all refinement schedules significantly outperform “No refine’’, and the trade-off pattern between H$\to$R$\to$SH and R$\to$H$\to$SH remains consistent, with the latter yielding higher reward at the cost of higher violations.
>
> Thank you again for your constructive feedback and willingness to raise your score.

---

> > ### Author Response · Authors · 2025-11-27
> >
> > Dear Reviewer, thank you again for your feedback. We hope we have addressed your concerns in the updated submission, and we would appreciate it if you could consider updating your score at your convenience.

---

### Official Review · Reviewer_vTvx · 2025-11-01

**Soundness:** 3
**Presentation:** 4
**Contribution:** 2
**Rating:** 4
**Confidence:** 5

**Summary:**

This paper integrates several established techniques—flow-based density modeling conditioned on state, latent-space refinement, and feasibility value estimation via a reversed expectile objective—to form a constraint-free offline safe reinforcement learning framework.

**Strengths:**

The paper is carefully executed and technically sound, and there is systematic integration of well-established components. It demonstrates a deep and precise understanding of existing literature: every methodological decision (e.g., shaping the tail of the density, re-centering distributions, introducing diffusion-based regularization) is well-motivated and mathematically coherent. Each modification clearly reflects an awareness of what has worked in prior studies and why it should succeed here.

**Weaknesses:**

This strength in technical maturity also highlights the main limitation: the work feels more like a thorough engineering synthesis than a piece of  innovation. Each component—while justified and effective—has appeared in prior forms, and the novelty stems mainly from their particular arrangement and empirical tuning. The result is a reliable and well-executed method, but one that does not significantly advance our theoretical or methodological understanding of safe offline reinforcement learning.

**Questions:**

1. Given the lack of a clean and standardized benchmark for flow-based models, and given the paper's  strong technical background, could this work have instead been directed towards something more like a systematic evaluation framework? This would be similar in spirit to, e.g., Clean Diffusion (NeurIPS 2023).

2. The paper integrates several known ideas—latent action manifold learning, flow-based density modeling, and lightweight latent refinement—into a new safe offline RL framework. While the individual components are not novel per se, their combination can still represent a meaningful contribution if the paper clearly articulates what new properties or advantages this combination introduces, such as PLAS (Zhou et al., 2021), Let Offline RL Flow (Akimov et al., 2022), and later latent-space safe RL methods. These studies should be explicitly acknowledged, as they form the conceptual foundation upon which this paper is built. Then, can the authors make any distinguishing features more clear (other than what I have acknowledged above in terms of strengths)?

3. The paper adopts a hard-constraint view of safety, similar to FISOR, where violations are strictly avoided. However, another common line of work treats cost as a budgeted resource—aiming to use the safety budget efficiently rather than minimizing it to zero. e.g. Safe Offline Reinforcement Learning with Real‑Time Budget Constraints (Lin et al., 2023). Acknowledging this alternative perspective would clarify that different methods reflect different design philosophies rather than differences in capability.

4. The density term (Eq. 2) plays an important role in constraining the policy within data-supported regions, but it is relatively under-explained and/or not clearly explained in the main text. I suggest adding a brief derivation or clarification in the appendix (e.g., how the flow-based likelihood or Jacobian term is computed and used, or a direct line between the bijective nature of the conditional probability and the resulting change-of-variables). This would improve clarity, especially for readers not familiar with conditional flow density modeling, and would make the paper more self-contained..

5. The GitHub repository lacks clear installation and setup instructions, which limits reproducibility. Please include dependency and compilation details (e.g., for the BulletSafetyGem component). Also, as part of the implementation appears inspired by FISOR, it would be helpful to note this explicitly in the documentation.

---

> ### Author Response · Authors · 2025-11-21
> **Response to the reviewer vTvx's comments (Part one)**
>
> **Q1: Could this be a system/benchmark paper?**
>
> Thank you for the suggestion. In our humble opinion, our work focuses mainly on proposing a new safe offline RL algorithm. Apart from proposing a new safe offline reinforcement learning method, we also present an explicit, low-cost control of distribution shift/OOD risk under a flow-based model, together with an optimization protocol (ordered small-step refiners) that makes this control operational. Turning the work into a full benchmark would shift its core contribution. That said, we agree that stronger standardization will materially improve the paper. In the revision, we have added exactly three concrete items:
> 1. **Computation resources.**
>    We have reported the computational cost of our model, including wall-clock time, memory, and FLOPs/step for training and inference in Appendix D.6.
>
> 2. **Reproducible configs & scripts.**
>    We have updated the released code and added necessary instructions, so all reported runs can be reproduced verbatim.
>
> 3. **Compact comparison across generative families.**
>    We have included an extended discussion and a side-by-side table contrasting CVAE-based (e.g., PLAS (Zhou et al., 2021)), flow-based (e.g., CNF (Akimov et al., 2022)), and other generative-based latent-space RL models in Appendix B.1.
>
> **Q2: Comparison with prior work**
>
> We appreciate your observation that our framework leverages well-established ideas—latent action manifolds, flow-based density modeling, and lightweight latent updates. Our contribution is not a new primitive per se, but a distinct design that arises from integrating these elements in the context of safe offline RL, along four axes:
> 1. **Task framing and scope.**
>    Prior flow-based methods (e.g., Let Offline RL Flow (CNF)) do not target safe offline RL. Our method sits in the flow-based family but is specifically instantiated for safe offline RL, with an explicit base-space KL control and ordered latent refiners.
>
> 2. **Why flows versus CVAE/diffusion.**
>    All three families exploit a latent manifold, but flows are invertible with exact likelihoods. Retaining a Gaussian base allows us to monitor a base-space KL and propagate it to action/policy deviation (TV/W$_2$) and OOD mass. This yields a quantified, tunable notion of conservatism that VAE (ELBO, non-invertible) and diffusion (costly likelihood, multi-step sampling) do not provide at comparable cost.
>
> 3. **Beyond Let Offline RL Flow (CNF).**
>    CNF reduces OOD by making the flow’s base uniform-bounded and freezing the decoder, but it does not quantify or tune deviation. In our framework, we (a) keep a Gaussian base and derive an explicit OOD/shift bound from the base-KL; (b) make this theory actionable via a shared refiner that adds a Gaussian-base pull (a base-KL regularizer); and (c) perform density shaping on the Gaussian base (using the flow’s exact inverse) so updates are pulled by density rather than pushed back by hard projections. Together, this turns conservatism into measurable and controllable, while keeping policy search within empirically safe, high-density regions.
>
> 4. **Stabilizing multi-objective safety optimization.**
>    Safe offline RL couples reward, safety, and OOD control. Instead of a single entangled loss, we use ordered small-step refiners in the base space with a frozen decoder—Safety → Reward → Shared—so updates remain in-support and non-expansive. This protocol links safety and OOD suppression, exposes a clear trade-off handle, and avoids the instability of lumping all terms into one gradient.
>
> We have added explicit citations and discussions to PLAS, CNF, and other related latent-space safe RL methods in the introduction section and Appendix B.1 to clearly situate our work in this line of research.
>
> **Q3: Hard vs. soft constraint safety**
>
> We agree that safe reinforcement learning spans two design philosophies: hard/near-zero-violation safety (our setting, akin to FISOR) and soft/budgeted safety, which treats cost as an allocatable resource (e.g., Lin et al., 2023). We intentionally target the first category because many real-world deployment scenarios require strict feasibility, while budgeted methods are more suitable when small violations are acceptable. In the revised manuscript, we have added a brief paragraph in Appendix B.2 to explicitly contrast these formulations and link this to the preliminary section, so it is clear this reflects a difference in philosophy and use case, not capability.

---

> ### Author Response · Authors · 2025-11-21
> **Response to the reviewer vTvx's comments (Part two)**
>
> **Q4: Density term**
>
> We thank the reviewer for this suggestion and we have added a detailed, step-by-step derivation of the conditional flow density term and its Jacobian in Appendix C.1, explicitly showing how Eq. (2) follows from the change-of-variables formula and how it is implemented in the multi-layer flow used in our method.
>
> **Q5: Code**
>
> We appreciate this comment and have updated the GitHub repository to include clear installation and setup instructions, as well as an explicit note that parts of our implementation are adapted from and inspired by the publicly released FISOR codebase.

---

> > ### Comment · Reviewer_vTvx · 2025-11-26
> >
> > I acknowledge receipt of this rebuttal.
> >
> > I appreciate the thoroughness of this response, as well as the response to the other reviewers. The revised manuscript is indeed improved. The most insightful response, in my opinion, is the answer to Q2 above and the delineation across four axes/dimensions. I am not suggesting another re-write, but the description above seems to "pop" more and is more crystalized than what is in the text. Is it table 4 that now drives these points home? And B.1? Perhaps B.1 could include this "integration along 4 axes" discussion.
> >
> > I finally note that there are some similarities in and across several of the review(er)s. There is some very interesting work done here and to borrow the review criteria text directly, I absolutely "would not mind if paper is accepted".

---

> > > ### Author Response · Authors · 2025-11-27
> > > **Response to Reviewer vTvx**
> > >
> > > Thank you again for your thoughtful follow-up comment. We agree that our answer to Q2 helped clarify how FLRP differs from prior work, and we have now integrated this framing into the paper. Specifically, we revised the abstract and introduction to highlight our contributions more clearly, retained Table 4 in the main text to summarize the first three axes, and added an explicit pointer to Appendix B.1, where we now discuss all four axes in more detail. We hope these changes further clarify the positioning and contribution of our work.
> > >
> > > If you feel that these, together with our earlier revisions, materially improve the clarity and contribution of the paper compared with the original submission, we would be very grateful if you could reconsider the score. Thank you for your time.

---

### Author Response · Authors · 2025-11-21
**Overall revisions on the paper**

The authors thank all reviewers for their insightful comments and constructive feedback. We have substantially revised the paper to further clarify the problem setting, sharpen the contributions, and add targeted experiments. Major revisions—highlighted in blue in the manuscript—are summarized as follows:

1. **Introduction and related work**: We revised both the Introduction and related work sections and added an extended discussion in Appendix B.1, explicitly distinguishing elements adapted from prior work from the novel contributions of FLRP. We also positioned our method within the broader landscape of generative latent-space approaches, supported by a side-by-side comparison table (Table 4).

2. **Refiner loss revision and Additional theory derivation**: We corrected a typographical error in the refiner losses and included a detailed derivation of Eq. (2) in Appendix C.1. We derived explicit reward and cost policy gap / performance bounds (Proposition 2 and Eqs. (P1)–(P4) in Appendix C.8) and added addtional discusions.

3. **Constraint formulation discussion**: We expanded our discussion of hard- vs. soft-constraint settings in offline reinforcement learning to clarify their trade-offs and implications.

4. **New ablation studies and discussions**:
   - (i) *Refiner order ablation* including reversed, random order, and no refiner approach. (Section 5),
   - (ii) *Reversed expectile parameter $\tau_h$ ablation* (Appendix E),
   - (iii) *Refiner loss weights ablation* (Appendix E),
   - (iv) *Optimization strategies*: comparing three decoupled refiners, a single unified refiner, and averaged updates (Appendix E),
   - (v) *Decoder update schemes*: frozen vs. alternating updates (Appendix E).

   These ablations demonstrate that FLRP is robust to a range of design choices, while the default configuration consistently achieves the best reward–cost balance.

5. **Computational analysis**: We added a discussion of training/inference costs and flow architectural choices in Appendix D, including a direct comparison between the flow prior and a Gaussian prior (Table 7).

6. **Presentation improvements**: We restructured Algorithm 1 into a clearer two-stage format and updated Figure 2 and other plots with improved labels and captions for better readability.

Due to space constraints, many of these updates are placed in the appendix. However, we are glad to make further revisions based on the reviewers’ ongoing feedback and suggestions.

---

### Meta-Review · Area_Chair_qSxc · 2025-12-05

**Summary:**

This paper proposes FLRP (Flow-guided Latent Refiner Policies), a safe offline RL framework that addresses hard safety constraints and out-of-distribution (OOD) action issues.  It integrates several established techniques: flow-based density modeling conditioned on state, latent-space refinement, and feasibility value estimation via a reversed expectile objective—to form a constraint-free offline safe reinforcement learning framework. Many designs in the algorithm are similar to FISOR (2024), and can be perceived as replacing the diffusion policy in FISOR with a normalizing flow policy and adding another post refiner stage.

The primary concerns of reviewers are the unclear novelty, complexity of the algorithm, as well as insufficient discussion. During the rebuttal, two reviewers increased (or considered increasing) their scores to 6, but one reviewer still holds a negative opinion on this paper. Weighing all the pros and cons of this paper, I think the paper could benefit from another round of careful refinement before being accepted in a conference like ICLR.

**Reviewer Concerns:**

Some concerns that I think are still not well-addressed:
- The work feels more like an engineering synthesis/combination of multiple known ideas, than a piece of innovation. (Reviewer vTvx, sYAh, 89xo, Zjrt) Despite the authors' rebuttal, I have read the paper and share a similar feeling.
- Too many hyperparameters to tune. (Reviewer sYAh)
- Computation complexity. (Reviewer sYAh, 89xo)
- Unclear role of refiner (Reviewer Zjrt)

**Reviewer Scores:**

Reviewer vTvx, 89xo, and sYAh increased (or considered increasing)  their scores to 6 during the rebuttal.

Reviewer Zjrt remains unsatisfied after the authors' rebuttal, and might not change his/her score.

---

### Decision · Program_Chairs · 2026-01-26

Reject